# LLM-Guided Communication for Cooperative Multi-Agent Reinforcement Learning

## Abstract

Communication can be essential in cooperative multi-agent reinforcement learning (MARL), where agents may need to overcome partial observability by exchanging information to accomplish tasks. However, prior methods often rely on messages that are uninterpretable or contain irrelevant information. To overcome this issue, we propose LLM-driven Multi-Agent Communication (LMAC), a novel MARL framework that combines LLM-based communication protocol design with a meta-cognitive latent representation module. LMAC employs iterative refinement with phase-specific feedback to produce interpretable protocols that enhance state recovery and shared understanding, while its latent module incorporates reliability signals with cycle consistency to ensure compact and trustworthy representations. Experiments across diverse MARL benchmarks demonstrate that LMAC consistently improves performance over other communication baselines.

## 1 Introduction

Cooperative multi-agent reinforcement learning (MARL) has emerged as a key paradigm for solving tasks where multiple agents must collaborate, such as autonomous driving (Chen et al., 2023a), network management, and strategic games (Nguyen et al., 2020; Orr & Dutta, 2023). In such environments, each agent learns from its own local observations, and partial observability prevents any single agent from fully reconstructing the global state required for effective decision-making (Zhu et al., 2024b). To address this, the centralized training with decentralized execution (CTDE) paradigm (Oliehoek et al., 2008) has been widely adopted, where centralized training leverages global information but execution remains decentralized. Within CTDE, value decomposition methods have been extensively studied to ensure proper credit assignment from the global value to individual utilities. Representative approaches include VDN (Sunehag et al., 2017), which expresses the joint value as a weighted sum of individual values, and QMIX (Rashid et al., 2018), which enforces the individual-global-max (IGM) condition through a mixing network.

Communication-based MARL allows agents to exchange information beyond their limited observations (Zhu et al., 2024a). Prior methods include sharing raw or compressed observations (Sukhbaatar et al., 2016; Das et al., 2019b; Li et al., 2022b) or exchanging structured representations such as agent influence, external knowledge, or global summaries (Wang et al., 2020; Du et al., 2022; Liu et al., 2024). However, latent-based messages in existing MARL approaches are often hard to interpret and may include redundant or missing task-critical information. Recent work has explored natural language mapping (Li et al., 2024), but this remains confined to simple tasks and largely imitates LLM agents rather than ensuring balanced situation awareness. Consequently, the same message may still be understood differently by agents, causing cooperation failures. For instance, in soccer, one player's call to "pass back!" may be understood differently by teammates, leading to miscoordination. Likewise, in MARL, inconsistent interpretation of the same message undermines cooperation, highlighting the need for protocols that are both interpretable and ensure consistent state understanding.

To overcome these limitations, we present LLM-driven Multi-Agent Communication (LMAC), a communication-based MARL framework that (i) designs interpretable protocols through LLM reasoning and (ii) learns compact meta-cognitive latent representations that exploit these messages. For protocol design, we leverage large language models such as GPT (OpenAI, 2023), Gemini (Gemini Team & Google, 2023), and Claude (Anthropic, 2024), which provide general-purpose and strong

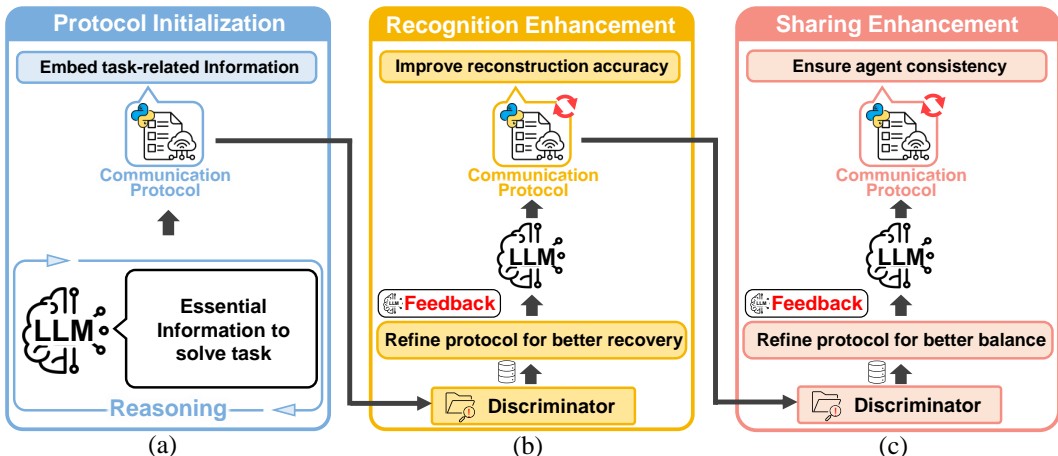

Figure 1: Illustration of protocol refinement in LMAC: (a) Protocol Initialization ($k = 0$) generates an initial protocol for sharing task-relevant information using their local observations, (b) Recognition Enhancement ($k = 1$) improves the accuracy of information recovery, and (c) Sharing Enhancement ($k = 2$) reduces inter-agent inconsistencies to ensure consistent understanding. Each stage refines the protocol under a distinct objective guided by discriminator feedback.

reasoning capabilities well-suited for developing communication protocols. Building on the Reflexion mechanism(Shinn et al., 2023), originally proposed for iterative self-refinement, we introduce phase-specific objectives where discriminators provide targeted feedback using real transitions. As shown in Fig. 1, refinement proceeds in three stages: (a) *Protocol Initialization*, where the LLM proposes a preliminary protocol; (b) *Recognition Enhancement*, where feedback improves information recovery; and (c) *Sharing Enhancement*, where feedback reduces inter-agent inconsistencies. This iterative, phase-specific process, with each stage concentrating on its own objective, yields protocols that capture task-relevant information, strengthen global awareness, and provide a shared cognitive basis for cooperation, enabling more interpretable and effective MARL.

To integrate these protocols into MARL training, we introduce a meta-cognitive latent module that reconstructs states with dimension-wise recovery signals and applies cycle-consistency to retain only task-relevant information. Combined with protocol design, this yields interpretable and consistent communication for effective cooperation. We validate our approach on multiple MARL benchmarks and show that it outperforms existing methods. Our contributions are summarized as follows:

- **LLM-based communication protocol design:** We propose an iterative Reflexion-inspired framework with phase-specific feedback and discriminators, yielding interpretable protocols that progressively enhance state recovery and mitigate imbalance.
- **Meta-cognitive representation learning:** We embed the designed protocols into MARL framework via a latent module that reconstructs states with dimension-wise recovery signals and enforces cycle consistency, ensuring messages are compact and reliably utilized.
- **Empirical evaluation and analysis:** We validate LMAC across diverse MARL benchmarks and provide in-depth trajectory analyses, showing how the designed protocols yield interpretable messages that directly enhance information recovery, consistency, and cooperative performance.

## 2 BACKGROUND

### 2.1 DEC-POMDPS WITH COMMUNICATION UNDER THE CTDE PARADIGM

Cooperative multi-agent reinforcement learning (MARL) with communication can be formalized as a decentralized partially observable Markov decision process with communication (Comm-Dec-POMDP), $G = \langle S, A, P, R, O, \mathcal{O}, I, n, \gamma, \mathcal{M} \rangle$. Here, $S$ is the global state space, $A$ the joint action space, $P$ the transition dynamics, $R$ the reward function, $O$ the observation function with observation space $\mathcal{O}$, $I$ the set of $n$ agents, $\gamma$ the discount factor, and $\mathcal{M}$ the message space. At timestep $t$, agent $i$ receives an observation $o_t^i = O_i(s_t)$ and selects an action $a_t^i$ through a decentralized

policy $\pi^i(\cdot \mid \tau_t^i)$ based on its trajectory $\tau_t^i := (o_0^i, a_0^i, \ldots, o_t^i)$. The objective is to maximize the expected cumulative reward $\mathbb{E}[\sum_{t=0}^{T-1} r_t]$, typically trained under the CTDE paradigm, where global information is available during training but only local observations are used at execution. As a baseline, QMIX (Rashid et al., 2018) learns a global action-value function $Q_{\text{tot}}(\tau_t, \mathbf{a}_t)$ under the IGM condition, ensuring consistency between global maximization and individual action-value function $Q^i(\tau_t^i, a_t^i)$. A key challenge in MARL is partial observability, since each agent only perceives a limited and noisy view of the environment.

To mitigate this issue, recent communication-based MARL methods (Goldman & Zilberstein, 2008; Foerster et al., 2016) are often equipped with a communication mechanism that allows them to exchange messages $m_t^i \in \mathcal{M}$ at each timestep. When such messages are incorporated, the value functions and policies are extended as $Q^i(\tau_t^i, m_t^i, a_t^i)$ and $\pi^i(\cdot \mid \tau_t^i, m_t^i)$. This formulation enables agents to leverage shared information to improve coordination and reduce uncertainty, ultimately enhancing cooperative performance in decentralized environments.

## 2.2 LARGE LANGUAGE MODELS FOR REASONING

Large language models (LLMs) such as GPT (OpenAI, 2023), Gemini (Gemini Team & Google, 2023), LLaMA (Touvron et al., 2023), and Claude (Anthropic, 2024) have rapidly expanded beyond natural language processing to domains requiring complex reasoning. Built on the Transformer architecture (Vaswani et al., 2017) with billions of parameters and trained on terabytes of text, these models exhibit strong capabilities in generation, contextual understanding, and abstract inference. To strengthen reasoning, methods such as Chain-of-Thought (CoT)(Wei et al., 2022) and Reflexion(Shinn et al., 2023) introduce intermediate steps and iterative refinement. In CoT, the model first produces reasoning tokens $z \sim f_\theta^{\text{LLM}}(x)$ from input prompt $x$ and then generates the final answer $y \sim f_\theta^{\text{LLM}}(x, z)$, where $f_\theta^{\text{LLM}}$ denotes an LLM with parameters $\theta$. Reflexion extends this process with feedback-driven updates: at step $k$, the model outputs $y^{(k)} \sim f_\theta^{\text{LLM}}(x, z^{(k)})$, derives a feedback sentence $c^{(k+1)} \sim f_\theta^{\text{LLM}}(x, \tilde{x}, z^{(k)}, y^{(k)})$ where $\tilde{x}$ is the feedback instruction, and then refines the reasoning as $z^{(k+1)} \sim f_\theta^{\text{LLM}}(x, c^{(k+1)})$, leading to $y^{(k+1)} \sim f_\theta^{\text{LLM}}(x, z^{(k+1)})$. This iterative procedure enables LLMs to revise their reasoning sequences based on prior errors and self-generated feedback, thereby achieving more consistent and robust performance in long-horizon problem solving. In this work, we leverage LLMs for communication protocol design in MARL, where Reflexion-based refinement is employed to progressively enhance the protocols.

## 3 RELATED WORKS

**Communication for MARL** In communication-based MARL, extensive studies have explored how agents can cooperate under partial observability through learned communication protocols (Sukhbaatar et al., 2016; Foerster et al., 2016). Research on what to communicate ranges from continuous messages in CommNet to efficient variants such as TMC (Das et al., 2019a), NDQ (Wang et al., 2020), and MAIC (Du et al., 2022), with later work addressing robustness to noisy channels (Zhang et al., 2019; Freed et al., 2020). Studies on when and with whom to communicate introduce gating and scheduling strategies (Singh et al., 2019; Karten et al., 2022; Niu et al., 2021; Xue et al., 2022; Yuan et al., 2023b; Hu et al., 2024). Finally, how to use messages has been studied through integration mechanisms such as attention in TarMAC (Das et al., 2019b) and representation learning in MASIA (Li et al., 2022b). While these methods improve coordination, the exchanged signals are typically uninterpretable and may not guarantee consistent recovery of task-relevant information across agents.

**Large Language Models for Reasoning** LLMs, pretrained on massive corpora, have demonstrated strong reasoning abilities beyond language generation. Chain-of-Thought prompting (Wei et al., 2022) has been extended into more structured reasoning formats (Zhou et al., 2022; Yao et al., 2023; Besta et al., 2023; Sel et al., 2023; Zhou et al., 2024), while zero-shot reasoning (Kojima et al., 2022) and instruction tuning with self-generated data (Wang et al., 2022) highlight the versatility of prompting. ReAct (Yao et al., 2022) integrates reasoning traces with environment interactions, and iterative refinement methods such as Reflexion (Shinn et al., 2023), Retroformer (Chen et al., 2024), and Expel (Zhao et al., 2024) enable continual self-correction. These advances establish LLMs as higher-level reasoning engines capable of stepwise abstraction and iterative improvement.

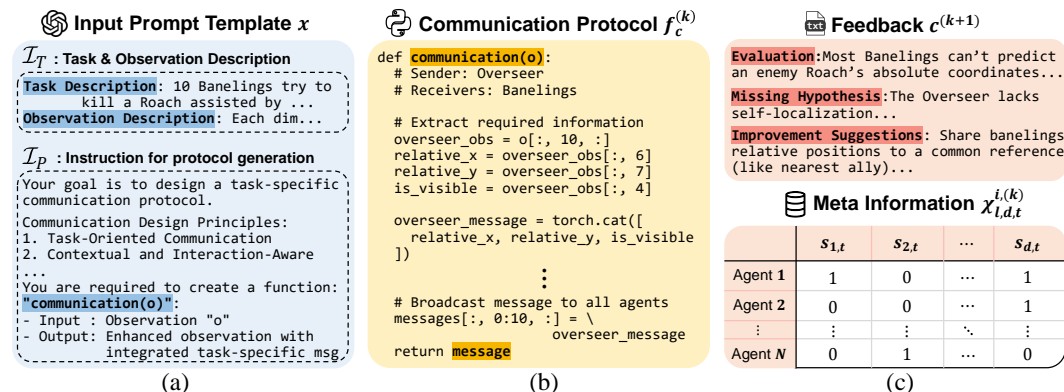

Figure 2: Elements of the LLM-based communication protocol refinement process: (a) Input prompt $x$ with task description $\mathcal{I}_T$ and design instruction $\mathcal{I}_P$, (b) generated protocol $f_C^{(k)}$ that maps local observations to agent-specific messages, and (c) meta-information data with phase-specific feedback instructions, which guide refinement by analyzing recovery accuracy and inter-agent imbalance.

**Large Language Models for Reinforcement Learning** Recent studies have applied LLMs to various aspects of reinforcement learning (Cao et al., 2025). Applications include reward design (Yu et al., 2024; Adeniji et al., 2023; Chu et al., 2023; Nair et al., 2022; Ma et al., 2024), trajectory summarization and task transformation (Du et al., 2023; Yuan et al., 2023a; Qiu et al., 2024), and state representation (Chen et al., 2023b; Wang et al., 2024; Da et al., 2024). LLMs have also been used directly as policy networks (Li et al., 2022a; Zitkovich et al., 2023; Shi et al., 2024), or for grounding actions into affordances and coordination policies (Ahn et al., 2022; Hu & Sadigh, 2023). More recently, their role in multi-agent cooperation has begun to be explored (Li et al., 2024; 2025; Agashe et al., 2025), though challenges remain in ensuring interpretable and consistent communication across agents.

Distinct from prior studies, our work leverages LLMs to design and refine communication protocols, explicitly aiming for interpretability, consistent state recovery, and stable cooperation in MARL.

## 4 METHODOLOGY

In this section, we provide the detailed description of our approach, expanding on the LLM-based communication protocol design and meta-cognitive representation learning introduced in Section 1.

### 4.1 LLM-GUIDED MULTI-PHASE COMMUNICATION PROTOCOL DESIGN

We now present a detailed description of our communication design. At each phase $k$, the LLM generates a communication protocol $f_C^{(k)}$ using the task prompt $x$ and reasoning tokens $z^{(k)}$, i.e.,

$$f_C^{(k)} \sim f_\theta^{\text{LLM}}(x, z^{(k)}), \ k \in \{0, 1, 2\} \tag{1}$$

where $f_C^{(k)}$ maps the agents' observation histories $(\tau_t^0, \dots, \tau_t^{n-1})$ to *interpretable messages* $(m_t^{0,(k)}, \dots, m_t^{n-1,(k)})$ for all agents. In other words, $f_C^{(k)}$ is designed to generate messages that explicitly correspond to and compose from dimensions of the agents' observations described in $x$, so that both agents and human can interpret what information is being exchanged and how it contributes to reconstructing task-relevant states. The refinement process at phase $k$ is then guided by feedback $c^{(k+1)}$, discriminator $D^{(k)}$, and feedback instruction $\tilde{x}^{(k+1)}$, which determine how the next protocol $f_C^{(k+1)}$ is constructed. Specifically, the update process is given by

$$c^{(k+1)} \sim f_\theta^{\text{LLM}}(x, \tilde{x}^{(k+1)}, D^{(k)}, f_C^{(k)}, z^{(k)}) \ \rightarrow \ z^{(k+1)} \sim f_\theta^{\text{LLM}}(x, c^{(k+1)}) \ \rightarrow \ f_C^{(k+1)} \sim f_\theta^{\text{LLM}}(x, z^{(k+1)}). \tag{2}$$

Here, $\tilde{x}^{(k+1)}$ is the phase-specific feedback instruction, $D^{(k)}$ is the discriminator evaluating how well $f_C^{(k)}$ meets the refinement goal, and $c^{(k+1)}$ is the feedback sentence used to update reasoning tokens. Unlike Reflexion, which applies a fixed feedback signal at every step, our framework

adaptively updates $\tilde{x}$ as objectives evolve across phases. Protocol design therefore proceeds in three stages, each with a distinct refinement goal:

**Protocol Initialization** ($k = 0$): The goal is to generate an initial communication protocol that enables agents to encode and exchange task-relevant information provided in the global state, reconstructing it as accurately as possible from their local observations and messages. To this end, we construct the task prompt $x = (\mathcal{I}_T, \mathcal{I}_P)$, where $\mathcal{I}_T$ specifies the task objectives and environment characteristics (state, observation, and action spaces), and $\mathcal{I}_P$ provides instructions ensuring that each agent $i$, given its history and message, can better infer the global state. The initial protocol $f_C^{(0)}$ is then generated accordingly.

The initial protocol $f_C^{(0)}$ is designed to embed task-relevant information for approximate state recovery, but it must be validated against actual environment trajectories. To this end, we employ a dataset $\mathcal{B}$ of sampled trajectories and define meta-information that measures how accurately each agent can recover state dimensions. Specifically, for agent $i$, state dimension $d$, and timestep $t$, we denote by $\chi_{l,d,t}^{i,(k)}$ the binary indicator of whether $s_{d,t}$ can be successfully reconstructed, where $l = 1$ represents the case with the message $m_t^{i,(k)}$ and $l = 0$ represents the case without it. Formally,

$$\chi_{l,d,t}^{i,(k)} = \mathbb{I}\left[\|\hat{s}_{l,d,t}^i - s_{d,t}\|^2 \leq \alpha\right], \tag{3}$$

where $\mathbb{I}$ is the indicator function, $\hat{s}_{1,d,t}^i = D_\phi^{(k)}(\tau_t^i, m_t^{i,(k)}, i)\big|_d$ (with message), and $\hat{s}_{0,d,t}^i = D_\phi^{(k)}(\tau_t^i, \mathbf{0}, i)\big|_d$ (without message). Here, $D_\phi^{(k)}$ is the discriminator parameterized by $\phi$ and trained to reconstruct the state using $\mathcal{B}$, while $\alpha$ denotes the reconstruction threshold. The temporal average $\mathbb{E}_t[\chi_{l,d,t}^{i,(k)}]$ then measures how well agent $i$ recovers dimension $d$ with or without messages, and the variance $\text{Var}_i[\chi_{l,d,t}^{i,(k)}]$ quantifies how unevenly this recovery is distributed across agents, serving as an indicator of information imbalance.

We then use these statistics to drive the refinement process in the subsequent phases as follows:

**Recognition Enhancement Phase** ($k = 1$): The goal of this phase is to improve recovery accuracy for each agent. To this end, the temporal average $\mathbb{E}_t[\chi_{l,d,t}^{i,(0)}]$ is obtained to identify cases where messages fail to support accurate recovery. Based on this data, the feedback instruction $\tilde{x}^{(1)}$ specifies why accuracy is lacking and how the protocol should be revised, guiding the LLM to refine $f_C^{(1)}$ toward encoding more task-relevant information.

**Sharing Enhancement Phase** ($k = 2$): Although $f_C^{(1)}$ improves recovery, inconsistencies may remain across agents. This phase exploits the variance $\text{Var}_i[\chi_{l,d,t}^{i,(1)}]$ to analyze why imbalance arises, such as when certain agents cannot identify specific state dimensions. From this data, the feedback instruction $\tilde{x}^{(2)}$ proposes concrete modifications to reduce heterogeneity, enabling the LLM to refine $f_C^{(2)}$ so that all agents consistently interpret shared information.

Through this process, we obtain the final communication protocol $f_C = (f_C^{(0)}, f_C^{(1)}, f_C^{(2)})$ that first initializes a protocol, then improves recognition, and finally reduces imbalance across agents. To illustrate the process, Fig.2 shows the overall refinement pipeline, including the task prompt, an example of the generated protocol, and the meta-information with feedback used for updating. Fig.3 presents phase-wise evaluation results, where the average meta-information $\mathbb{E}_t[\chi_{1,d,t}^{i,(k)}]$ increases steadily, indicating improved information recovery, while the variance across agents $\text{Var}_i[\chi_{1,d,t}^{i,(k)}]$ decreases, reflecting reduced information imbalance. These trends show that as refinement progresses, the designed protocol enhances information recognition, alleviates information imbalance among agents, confirming that the framework achieves its intended design. Additional details on prompt construction, discriminator training, and trajectory dataset preparation are provided in Appendix B.

## 4.2 META-COGNITIVE REPRESENTATION LEARNING FOR MARL FRAMEWORK

Based on the final communication protocol $f_C$, we define each agent's aggregated message as $m_t^i = \left(m_t^{i,(0)}, m_t^{i,(1)}, m_t^{i,(2)}\right) = f_C(\boldsymbol{\tau}_t)$. Using these messages, we propose LMAC, a MARL framework

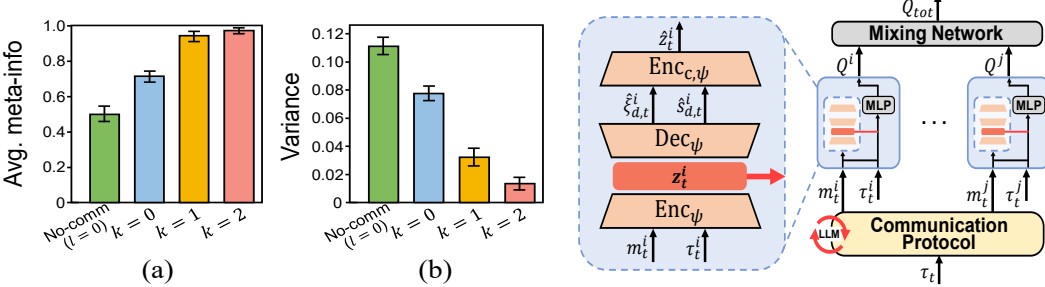

Figure 3: Phase-wise evaluation of meta-information: (a) Average values and (b) variance across agents

Figure 4: Overall framework of the proposed LMAC

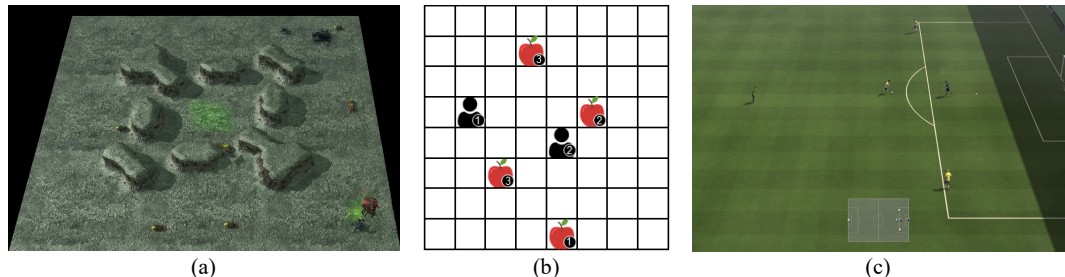

Figure 5: MALR Benchmarks used in our experiments: (a) SMAC-Comm, (b) LBF, and (c) GRF

that integrates LLM-driven communication into CTDE training. We adopt QMIX (Rashid et al., 2018) as the baseline, aiming to provide each individual $Q^i$ with informative representations of the global state, though the approach can also generalize to general CTDE methods such as VDN (Sunehag et al., 2017) and QPLEX (Wang et al., 2021). Rather than feeding raw messages, we employ an encoder–decoder $(\text{Enc}_\psi, \text{Dec}_\psi)$ with parameter $\psi$ to compress and reconstruct information, addressing the inefficiency of high-dimensional states and the redundancy of raw messages. The latent representation is defined as $z_t^i = \text{Enc}_\psi(\tau_t^i, m_t^i)$, and the encoder–decoder is trained to reconstruct both the state $s_t$ and auxiliary meta-information $\xi_{d,t}^{\cdot,i} = \mathbb{I}[\|\hat{s}_{d,t}^i - s_{d,t}\|^2 \le \alpha]$, where $\hat{s}_{d,t}^i$ is the reconstructed $d$-th state dimension. Here, $\xi$ reuses the meta-information idea from protocol design, allowing agents to identify which state dimensions are accurately captured and which remain uncertain.

In addition, to prevent irrelevant information from being encoded, we adopt a cycle-consistency loss inspired by Zhu et al. (2017). Specifically, the latent $z_t$ is decoded and then re-encoded using an auxiliary encoder $\text{Enc}_{c,\psi}$, enforcing $\hat{z}_t \sim \text{Enc}_{c,\psi}\big(\text{Dec}_\psi(z_t)\big)$ so that $\text{Enc}_{c,\psi}$ is trained to reconstruct $z_t$. The key intuition is that any redundant information in $z_t$ is discarded during decoding and thus cannot be recovered by $\text{Enc}_{c,\psi}$. As a result, the model learns to encode only reconstructable, task-relevant features while suppressing noise. Finally, the learned latent $z_t^i$ is incorporated into the individual utilities $Q^i(\tau_t^i, z_t^i)$, and the joint action-value $Q^{\text{tot}}$ is optimized via TD-learning. The overall framework is shown in Fig.4, and further details, including algorithm, loss functions, and training procedures, are provided in AppendixB.

## 5 EXPERIMENTS

In this section, we evaluate the proposed method on three benchmark environments shown in Fig.5: StarCraft Multi-Agent Challenge with Communication (SMAC-Comm) (Samvelyan et al., 2019), a communication-intensive variant of StarCraft II evaluated on `bane_vs_hM`, `1o_10b_vs_1r`, `2o_20b_vs_2r`, and `5z_vs_1ul`; Level-Based Foraging (LBF) Papoudakis et al. (2021a), a cooperative foraging task with settings `8x8-2p-2f-s1-coop` and `11x11-6p-4f-s1-coop`, ($n \times n$: grid size, $p$: agents, $f$: fruits, $s$: sight range); and Google Research Football (GRF) (Kurach et al., 2020), a cooperative soccer game with scenarios `3_vs_1_with_keeper` and `run_pass_and_shoot`. We first compare performance against other communication baselines,

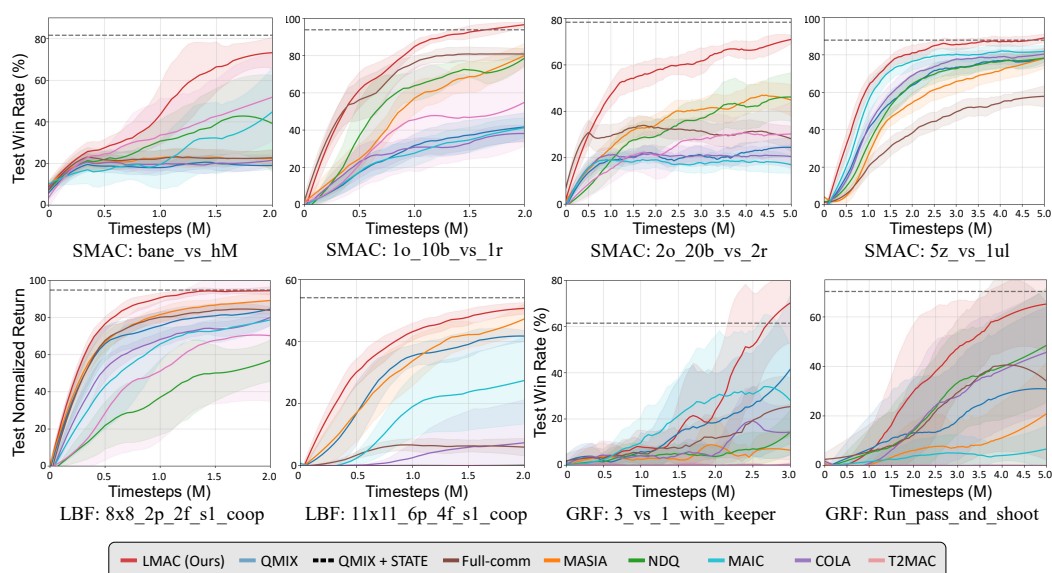

Figure 6: Performance comparison in various MARL benchmarks

then analyze how phase-specific protocols contribute to cooperation. Unless otherwise specified, experiments use `gpt-4.1-2025-04-14` as the backbone LLM, with other variants included in ablation studies. All results are averaged over 5 random seeds with standard deviations, and more experimental details are provided in Appendix C.

## 5.1 PERFORMANCE COMPARISON

To validate our approach, we compare LMAC against a broad set of communication-based MARL methods. The comparison includes our baseline **QMIX**(Rashid et al., 2018), **FullComm**, where agents broadcast full observations to all teammates, and **QMIX+State**, where each agent is provided with the global state as an upper-bound reference. We further evaluate against **NDQ**(Wang et al., 2020), which reduces communication cost via decomposable value functions; **MASIA**(Li et al., 2022b), which aggregates information through self-supervised representation learning; **MAIC**(Du et al., 2022), which generates incentive messages to bias teammates' utilities; **COLA**(Monda et al., 2023), which improves coordination through inter-agent consensus; and **T2MAC**(Liu et al., 2024), which enables selective communication via evidence-driven integration. All baselines are evaluated using author-released implementations. For our method, results are reported with the best-performing threshold $\alpha$, while full hyperparameter settings and further details of competing algorithms are provided in Appendix C.

Fig. 6 reports success rates across the three benchmark environments. On SMAC-Comm, our method achieves faster convergence and higher final success rates across all four scenarios, with particularly large gains on `bane_vs_hM` and the large-scale `2o_20b_vs_2r`, showing strong scalability when state recovery is difficult or agent numbers grow. Notably, our results nearly match the upper-bound QMIX+State, indicating that the designed protocols effectively capture the most critical state information. On LBF, similar trends appear: our method learns faster and consistently reaches higher final performance, again approaching QMIX+State. This confirms that interpretable and refined communication enables sufficient state reconstruction for coordinated behavior. On GRF, our method not only surpasses all baselines but even outperforms QMIX+State in final success rates. Since GRF involves high-dimensional observations, simply giving all agents the full state leads to excessive dimensionality and slow convergence. By contrast, our latent learning compresses messages into compact task-relevant features, enabling faster convergence and stronger cooperative strategies. This directly demonstrates the efficiency of our latent representation design. Overall, these results highlight that our framework consistently improves learning speed and final performance, while allowing agents to exploit information more effectively across diverse MARL settings. In addition, to verify

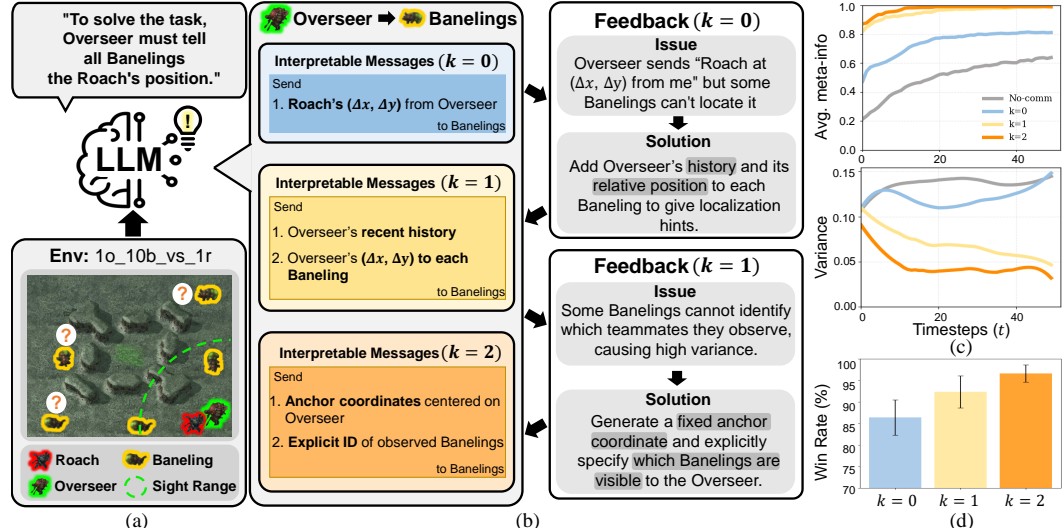

Figure 7: Protocol refinement analysis on SMAC 1o_10b_vs_1r: (a) Task scenario with Overseer, Roach and Banelings under partial observability, (b) protocol messages and corresponding feedback at each phase $k$, (c) trajectory-level averages and variances of meta-information with ($k = 0, 1, 2$) or without messages (No-comm), and (d) average win rates across phases.

the generality of our approach, we also apply LMAC to VDN Sunehag et al. (2017) and QPLEX Wang et al. (2021) in Appendix E and observe similarly significant performance gains.

## 5.2 TRAJECTORY ANALYSIS

To analyze how our framework yields interpretable communication protocols that improve information recovery and balance, we conduct a trajectory analysis on the SMAC 1o_10b_vs_1r map, summarized in Fig.7. In (a), the task requires 10 Banelings (10b) to quickly converge on a Roach (1r), with the Overseer (1o) providing positional cues. Because absolute positions are not included in raw observations, the LLM reasons that agents must infer both the Roach's location and each Baneling's absolute position; otherwise, delays occur and less damage is dealt. In (b), protocol evolution is shown: at $k = 0$, the Overseer broadcasts "the Roach is $\Delta x, \Delta y$ (relative position) away from the Overseer," enabling partial localization but failing without knowledge of the Overseer's position. At $k = 1$, feedback notes that "the Overseer's position is difficult to identify," so the protocol is refined to include the Overseer's relative position and recent history as localization hints. At $k = 2$, variance-based feedback highlights that "some Banelings still cannot identify which teammates they observe," prompting the use of a fixed anchor coordinate centered on the Overseer with explicit IDs of observed agents. The final protocol thus shares the Roach's relative position, the anchor, and teammate IDs, allowing consistent absolute recovery. In (c), average meta-information increases in phase 1, while its variance across agents drops in phase 2. In (d), learning performance improves across phases as agents achieve shared localization and coordinate attacks. These results confirm that our method yields interpretable messages whose refinements directly enhance recovery and eliminate imbalance, validating the intended protocol design. Similar patterns are observed in other environments, with additional analyses provided in Appendix E.

## 5.3 ABLATION STUDY

We conduct three ablation studies in the SMAC-Comm environment to validate the contributions of our framework: (a) component evaluation, (b) comparison of different LLM variants, and (c) the effect of the reconstruction threshold $\alpha$. We additionally conducted experiments on the effect of the number of update phases, the impact of reduced communication capacity, and computational complexity, which can be found in Appendix E, F.

**Component Evaluation:** To analyze the effect of each component, we compare the performance of protocols obtained at different refinement stages ($k = 0, 1, 2$), as well as two variants of our framework: one without the consistency loss ('w/o Consistency') and one without meta-information

Table 1: Ablation study on SMAC-Comm: (a) Component evaluation (b) LLM variants (c) Comparison across reconstruction threshold $\alpha$

| Setting | Avg. Win Rate(%) |
|---|---|
| w/o Consistency | $66.5 \pm 2.1$ |
| w/o Meta | $76.6 \pm 5.6$ |
| k = 0 | $68.5 \pm 3.8$ |
| k = 1 | $77.8 \pm 2.2$ |
| k = 2 (Ours) | $\mathbf{82.9 \pm 1.9}$ |

(a) Component evaluation

| LLM | Avg. Win Rate(%) |
|---|---|
| GPT | $\mathbf{82.9 \pm 1.9}$ |
| GPT-mini | $79.8 \pm 1.5$ |
| GPT-o1-mini | $81.8 \pm 2.9$ |
| Claude | $81.9 \pm 2.6$ |
| Gemini | $80.8 \pm 1.6$ |

(b) LLM variants

| $\alpha$ | Avg. Win Rate(%) |
|---|---|
| 0.0005 | $77.2 \pm 2.1$ |
| 0.002 | $79.3 \pm 2.8$ |
| 0.005 | $80.5 \pm 1.7$ |
| 0.05 | $\mathbf{82.9 \pm 1.9}$ |
| 0.5 | $80.2 \pm 3.2$ |

(c) $\alpha$ comparison

reconstruction ('w/o Meta'). As shown in Table 1(a), performance steadily improves as the refinement phase progresses, consistent with our earlier findings that state recovery and balance improve with each stage. Removing the consistency loss causes a clear drop in performance, demonstrating that eliminating redundant information in messages is crucial for learning. Similarly, removing meta-information also degrades performance, confirming that knowing not only the recovered state but also its reliability is essential for effective cooperation. Overall, these results show that every component of our framework contributes substantially to performance.

**LLM Variants:** Table 1(b) evaluates our method with different LLMs, including GPT-4.1, GPT-4.1-mini, o1-mini, Gemini-2.5-Flash, and Claude-Opus. Results show that all recent LLMs achieve strong performance, with GPT providing the highest success rates. However, smaller and more efficient models (e.g., GPT-4.1-mini, o1-mini) still perform competitively, demonstrating that the key driver of performance is the multi-phase refinement process rather than reliance on a particular model. This refinement procedure further encourages consistency in the resulting communication protocols, ensuring that our framework remains broadly applicable across different LLMs.

**Reconstruction Threshold $\alpha$:** Finally, Table 1(c) analyzes the effect of the reconstruction threshold $\alpha$ used in constructing meta-information. A too small $\alpha$ makes the criterion overly strict, causing most dimensions to be judged unrecoverable and leading to excessive message generation. While performance is maximized at $\alpha = 0.05$, results remain stable for larger values, suggesting that our method is not highly sensitive to this parameter. This robustness indicates that our framework can be applied without heavy hyperparameter tuning.

## 6 LIMITATION

While our framework performs well and behaves as we intended, there remain a few limitations. First, the approach introduces some overhead from training the discriminator and interacting with LLMs during protocol refinement. However, as shown in the computational complexity analysis in Appendix F, the discriminator contributes only a small fraction to the total training cost, and the limited number of LLM queries per refinement phase keeps the overhead modest. Second, the protocol design partially depends on the reasoning ability of the chosen LLM. Nevertheless, as shown in our ablation study, the method achieves consistent performance across different LLM variants, and with the rapid progress in LLM reasoning, this dependency is unlikely to pose a major obstacle going forward.

## 7 CONCLUSION

In this work, we introduced LMAC, a new communication-based MARL framework that combines LLM-guided protocol refinement and meta-cognitive latent learning. Our method designs interpretable communication protocols through multi-phase refinement with discriminator feedback, ensuring that agents recover task-relevant information consistently and alleviate information imbalance. The refined messages are then integrated into MARL framework via a latent module that reconstructs states with reliability-aware meta-information, while cycle consistency enforces compactness. Through this design, LMAC provides communication that is both interpretable and effective, enabling agents to achieve shared cognition and more stable cooperation. Empirical results on SMAC-Comm, LBF, and GRF benchmarks show that our approach consistently achieves faster convergence and higher final performance than strong communication baselines.

## ETHICS STATEMENT

This work focuses exclusively on algorithmic contributions in cooperative multi-agent reinforcement learning (MARL) under the CTDE paradigm. All experiments were conducted using publicly available simulation environments, including SMAC-Comm (Samvelyan et al., 2019; Wang et al., 2020), LBF (Papoudakis et al., 2021a), and GRF (Kurach et al., 2020). These benchmarks do not involve human subjects, personal data, or sensitive information. LLMs were used only for generating and refining communication protocols within our proposed framework and for minor editorial polishing, without influencing research ideation, problem formulation, or analysis. We do not foresee harmful applications beyond the intended scope of cooperative agent research. All work was conducted in compliance with the ICLR Code of Ethics, with careful attention to fairness, transparency, and research integrity.

## REPRODUCIBILITY STATEMENT

We have made substantial efforts to ensure reproducibility of our results. Section 4 provides the detailed description of our proposed framework, and Appendix C includes the implementation details. The benchmarks used in our experiments are all publicly available, and code links are provided in Appendix C. An anonymized code repository containing our implementation and experiment scripts is submitted as supplementary material, with further explanations included in the implementation details. The baselines and their official code repositories are listed in Appendix C.2, ensuring that all results reported in this paper can be independently reproduced.

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

## A    THE USE OF LARGE LANGUAGE MODELS

In this work, LLMs were employed only for two purposes: (i) as part of our proposed framework, where API-based models (e.g., GPT-4.1-2025-04-14 and variants) were used to generate and iteratively refine communication protocols, and (ii) for polishing grammar and improving the clarity of our writing. LLMs were not involved in research ideation, problem formulation, method design, experimentation, analysis, or interpretation, and all scientific content and decisions were made by the authors.

## B    IMPLEMENTATION DETAILS

This section provides the implementation details of our refinement process. B.1 describes the construction of prompts used in different refinement phases. B.2 presents examples of LLM outputs for both protocol generation and feedback. Finally, B.3 details the training objectives of LMAC, including the reconstruction, meta, and consistency losses.

### B.1    PROMPT CONSTRUCTION FOR REFINEMENT PROCESS

**Task prompt** $x$**:** Here, $x$ is the phase-specific prompt designed to guide the generation of communication protocols, and the LLM outputs a Python code based implementation of the protocol. For each environment, we construct task descriptions based on the original scenarios defined by their respective authors, including SMAC-Comm (Samvelyan et al., 2019; Wang et al., 2020), LBF (Du et al., 2022), and GRF (Kurach et al., 2020). Furthermore, the mapping of each observation and state dimension into natural language is guided by Wang et al. (2024), and representative examples are illustrated below.

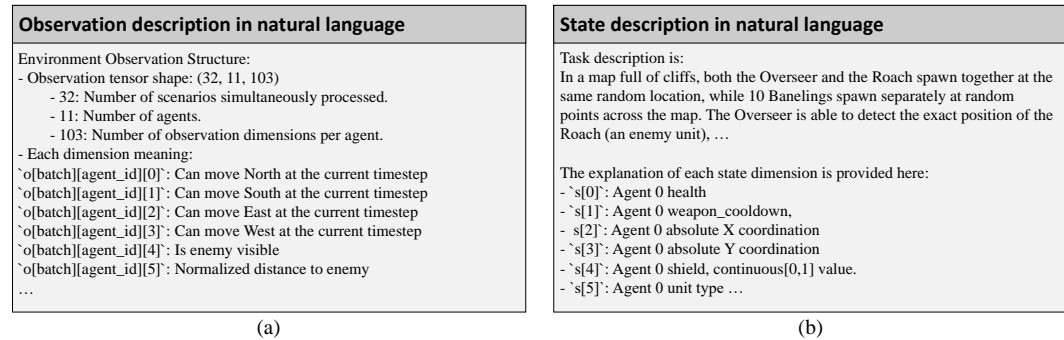

Figure B.1: (a) Example of observation description , (b) Example of state description

The task prompt is defined as $x = (\mathcal{I}_T, \mathcal{I}_P)$, where $\mathcal{I}_T$ specifies the task objectives and environment characteristics, and $\mathcal{I}_P$ provides explicit instructions for protocol generation. Before directly instructing the LLM to design a protocol, we first guide it through CoT (Wei et al., 2022) reasoning to identify task-critical state dimensions. This reasoning step filters out irrelevant dimensions in high-dimensional states and ensures that the protocol concentrates on essential information. In particular, $\mathcal{I}_T$ is constructed as a prompt describing the task and environment properties, while $\mathcal{I}_P$ provides detailed instructions for communication protocol design. Their respective templates are presented below.

**Task Description Template($I_T$)**

**Task Context - Task Objectives and Environment Characteristics:**
**Task Description:**
{task_description}
**Environment Observation Structure:**

- Observation tensor shape: {obs_shape}
- {obs_dim_desc}
- Each dimension meaning: {detail_content}

**Environment Characteristics:**
- Multi-agent partially observable environment
- Agents must coordinate under incomplete information
- Communication enables sharing of non-locally observable information

---

**Protocol Generation Template($I_P$)**

**Communication Design Key Principles**:

1. **Task-Oriented Communication**:
- Explicitly identify observation dimensions crucial to solving the task.
- Messages must clearly relate these dimensions to task objectives.

2. **Uniqueness, Sufficiency & Compactness**:
- Each agent should communicate information that others do not already possess or cannot easily infer based on their own observations.
- Communication should ensure sufficiency, meaning that agents exchange enough information to enable effective inference and coordination under partial observability.
- At the same time, messages should maintain compactness, minimizing redundancy and avoiding the transmission of unnecessary or easily inferable data.

3. **Contextual and Interaction-Aware**:
- Communication should be based on the agent's own observations, actively leveraging behavior-relevant information derived from its "perceived possibilities and recent behavior patterns".
- In environments where direct observation of allies and enemies is severely limited, agents should emphasize sharing self-perceived behavioral information, such as movement possibilities and recent actions, if available.

4. **Explicitness and Clarity**:
- Avoid overly abstract messages. All information critical for solving the task must be included explicitly, in a clear and interpretable form.

5. **Structured Output**:
- The final output should be a tensor of shape ({batch, agents, obs_dim} + message_dim).

6. **Communication Protocol**:
- Specify whether information is exchanged via peer-to-peer (agent-specific) or broadcast (global).
- Each agent should customize received messages based on the context and utility.
- Messages produced by an agent must be distributed and concatenated into the observations of other agents, never into only its own.
- Each message must include a sender identity field (one-hot encoded vector).

7. **Computational Efficiency**:
- No trainable components (e.g., neural networks) in the communication function.

**Observation Access Pattern**:
For example:

- o[2, 0, :] = observation vector of agent 0 in the 2nd batch
- o[2, 2, :] = observation vector of agent 2 in the 2nd batch

**Protocol Requirements**:
Using the important state dimensions reasoning Tokens above, design a communication protocol that enables agents to share information about these critical dimensions to improve coordination.
You are required to create two Python functions:
1. `message_design_instruction()`:
- Clearly describes how the message content is constructed based on the important state dimensions and observation context
2. `communication(o)`:
- Input: Observation tensor `o`
- Output: Enhanced observation tensor with integrated task-specific messages
- Focus on communicating information related to the important dimensions identified above
Both functions must be executable and ready for direct integration with MARL algorithms.
Caution!: Create two Python functions that minimizes the use of "for loops" when handling batch processing to optimize computational efficiency.
Let's think step by step. Below is an illustrative example of the expected output:

```python
import torch as th
def message_design_instruction():
#Explain how this protocol helps agents coordinate using these critical dimensions
return message_description
def communication(o):
#Your communication implementation focusing on important dimensions
#You should design the communication protocol based on the message design instruction
#use same device as input to avoid CUDA/CPU mismatch
return messages_o
```

**Feedback instruction:** $\tilde{x}$ is a prompt designed to generate natural language feedback by analyzing the limitations of the current protocol through meta-information derived from the discriminator's evaluation and by suggesting directions for improvement. The *Feedback Prompt Template* follows a fixed structure with the phase index, objectives, phase instruction, and discriminator results. Among these, the *phase instruction* provides the phase index with a clear goal and concise guidance on how the protocol should be refined. In the *Recognition Enhancement* phase it emphasizes improving reconstruction of critical state dimensions and addressing uneven prediction across agents, while in the *Sharing Enhancement* phase it focuses on reducing cross-agent imbalance and ensuring consistent state recovery through coordination cues and temporal signals.

---

**Feedback Prompt Template**

You are an analysis agent tasked with improving communication strategies in a multi-agent reinforcement learning (MARL) system.
{x}
============================================================
{phase_instruction}
============================================================
**Previous Protocol Under Analysis**:
{cur_communication_method}
============================================================
**DISCRIMINATOR EVALUATION RESULTS**:
{meta_information}
============================================================

**Analysis Context**:
- Each agent combines its own local observation with received messages to infer important state dimensions
- You are analyzing predictability results showing how accurately each agent can infer critical dimensions
- Performance differences across agents indicate areas where communication protocol needs improvement
========================================================
**Expected Output Format (JSON)**:
```
{
"Evaluation": "...",
"Missing_Information_Hypothesis": "...",
"Improvement_Suggestions": "..."
}
```

To update the protocol with the generated feedback, the following additional context template is appended to the protocol generation prompt.

### Additional context Template

Here is the feedback from the previous communication protocol evaluation:

{feedback}

Reflect the feedback by designing messages from each agent's own observations, prioritizing information that enables all agents to achieve consistent state prediction and shared understanding.

## B.2 Details of LLM Outputs for Protocol and Feedback

**Communication Protocol** Since protocol generation relies on high-dimensional tensors, the LLM may not align dimensions correctly in one attempt. We therefore adopt a Reflexion-based mechanism, where error messages are fed back for iterative correction. Expressing the protocol in code allows direct verification of transmitted information, while the accompanying message design instruction makes the communication process interpretable even to non-experts. Here is the example of the communication protocol.

---

**Example of Generated Protocol**

```python
import torch as th

def message_design_instruction():
    """
    Message Design Instruction:

    - **Sender**: Only the Overseer (agent at index 10) sends
        messages.
    - **Receivers**: All Banelings (agent indices 0~9) receive the
        message.
      The Overseer does not receive any messages.
    - **Message Content**:
        - `relative_x`: Overseer's observed relative X position of
            the enemy
          Roach (`o[..., 6]`).
        ...
    - **Why**: Only the Overseer can observe the Roach; Banelings
        need this
      precise and actionable info to coordinate and attack.
        Including sender
      identity ensures explicit grounding and prevents ambiguity.
        ...
    """
    return (
        "Message Structure: [relative_x, relative_y, is_visible, "
        ...
    )

def communication(o):
    device = o.device
    batch_size, n_agents, obs_dim = o.shape
    message_dim = 14

    # 1. Get Overseer's observation (agent index 10)
    overseer_obs = o[:, 10, :]  # (batch, 103)

    # 2. Extract required info from Overseer
    relative_x = overseer_obs[:, 6].unsqueeze(1)  # (batch, 1)
    relative_y = overseer_obs[:, 7].unsqueeze(1)  # (batch, 1)
    is_visible = overseer_obs[:, 4].unsqueeze(1)  # (batch, 1)
    ...
    # 5. Broadcast message to all agents
    messages = th.zeros(batch_size, n_agents, message_dim,
        device=device)
    messages[:, 0:10, :] = overseer_message.unsqueeze(1).expand(-1,
        10, -1)

    # 6. Concatenate messages to observations
    messages_o = th.cat([o, messages], dim=2)  # (batch, 11, 117)
    return messages_o
```

---

**Meta-information:**  Meta-information serves as a quantitative indicator of how communication improves state prediction compared to the no-message baseline. In the *Recognition Enhancement* phase, it measures the average success rate of reconstructing state dimensions, as can be seen in Figure B.2.(a), which shows the overall improvement achieved through communication. In the *Sharing Enhancement* phase, it captures the variance of prediction performance across agents over time, as illustrated in Figure B.2.(b), enabling a more fine-grained analysis of consistency.

<div>

**Meta-information $E_t\left[\chi_{l,d,t}^{i,(k)}\right]$**

```
"dimension": "s[62] – Agent 10 absolute X coordination",
    "with_communication": {
        "agent_success_rates": [0.64, 0.56, … , 0.65, 0.95]
    },
    "without_communication": {
        "agent_success_rates": [0.52, 0.47, … , 0.39, 0.94]

…
"dimension": "s[63] Agent 10 absolute Y coordination",
    "with_communication": {
        "agent_success_rates": [0.98, 0.98, … , 0.97, 1.0]
    },
    "without_communication": {
        "agent_success_rates": [0.38, 0.37, … , 0.39, 1.0]
…
```

</div>

<div>

**Meta-information $Var_i\left[\chi_{l,d,t}^{i,(k)}\right]$**

```
"dimensions":"s[62] – Agent 10 absolute X coordination",
    "with_communication": {
        "early": [0.407, … , 0.775], variance: [0.033],
        …
        "mid": [0.791, … , 0.978], variance: [0.013]
        …
        "late": [0.766, … ,1.0], variance: [0.015]
    },

    "without_communication": {
        "early": [0.219, … , 0.769], variance: [0.363],
        …
        "mid": [0.575, … , 0.981], variance: [0.243],
        …
        "late": [0.716, … , 1.0], variance: [0.143]
```

</div>

(a)                                    (b)

Figure B.2: (a) Example of temporal average $\mathbb{E}_t[\chi_{l,d,t}^{i,(k)}]$ based on meta-information , (b) Example of variance $\mathrm{Var}_i[\chi_{l,d,t}^{i,(k)}]$ based on meta-information

**Natural Language based Feedback:**  The natural language feedback $c^{(k+1)}$ transforms the quantitative meta-information into actionable and interpretable guidance that directly supports protocol refinement. It consists of three parts: (i) evaluation of the current protocol, (ii) hypotheses about missing information, and (iii) concrete suggestions for improvement. An example can be found below, which illustrates how quantitative results are translated into actionable refinements.

<div>

**Example natural language feedback $c^{(k+1)}$**

**Evaluation :** The current communication protocol enables the Banelings (agents 0–9) to receive Overseer's (agent 10) relative enemy position ... This significantly improves prediction accuracy of critical state dimensions ... compared to no communication. However, the success rates across Banelings are uneven and notably lower than the Overseer's own high accuracy, indicating inconsistent inference ...

**Missing information hypothesis :** The protocol currently misses explicit indicators of agent-specific observation reliability or visibility, which would clarify which agents have direct knowledge of the enemy or Overseer positions and which rely solely on communication. It also omits behavioral or temporal context that could help agents disambiguate relative positioning over time. ...

**Improvement suggestions :**
1. Include an explicit visibility or reliability flag per agent in the message to indicate whether the Overseer currently has direct, reliable observation of the enemy and itself, enabling agents to weigh communicated information appropriately.
2. Augment the message with behavioral cues such as a timestamp or sequence number to help agents track message freshness and temporal consistency.
...

</div>

### B.3 Training Losses of LMAC

**Discriminator Training:** The training dataset $\mathcal{B}$ consists of 5000 trajectories per environment, collected at 200k training steps under $\epsilon$-greedy exploration with the initial protocol fixed across five seeds to avoid bias and capture its limitations as a basis for refinement. Based on this dataset, the discriminator $D^{(k)}$ evaluates how well a candidate communication protocol $f_C^{(k)}$ contributes to state reconstruction: for each agent $i$, it takes the local observation $o_t^i$ and the message $m_t^{i,(k)}$ as input and reconstructs the global state $s_t$. The training objective is defined as

$$\phi^{*(k)} = \arg\min_{\phi} \mathbb{E}_{(o,m^{(k)},s)\sim\mathcal{B}} \|D_\phi^{(k)}(o_t^i, m_t^{i,(k)}) - s_t\|_2^2, \tag{B.1}$$

where $D^{(k)}$ is implemented as an autoencoder that compresses the input into a latent representation and reconstructs it to estimate the global state. Training is conducted with $\mathcal{B}$ using MSE loss and mini-batch SGD in a supervised learning setup. The reconstructed outputs are further used to compute the meta-information $\chi_{l,d,t}^{i,(k)}$, which provides recognition accuracy and imbalance indicators.

Table B.1: Hyperparameters used for training the discriminator $D^{(k)}$.

| Parameter | Value |
|---|---|
| Batch size | 32 |
| Dropout rate | 0.1 |
| Epochs | 1000 |
| Iterations per epoch | 10 |
| Hidden dimension | 64 |
| Latent dimension | 20 |
| Learning rate | 0.0005 |
| Optimizer | Adam |

**Representation and Policy Training:** The Meta-Cognitive Representation Learning module is implemented as an autoencoder architecture. The encoder $Enc_\psi$ incorporates an attention mechanism, where the query is formed from the current observation and received messages, while the key and value are derived from the trajectory $\tau_t^i$. This allows the latent representation $z_t^i$ to capture how the current observation and message attend to $\tau_t^i$. The decoder $Dec_\psi$ takes the latent representation as input and produces two outputs: (i) an estimate of the global state $\hat{s}_t^i$, and (ii) the meta information $\hat{\xi}_{d,t}^i$. The meta information is defined as $\xi_{d,t}^i = \mathbb{I}\left[\left(\hat{s}_{d,t}^i - s_{d,t}\right)^2 \leq \alpha\right]$, which indicates whether agent $i$, given its messages and trajectory, can accurately reconstruct a particular state dimension. The training objective for these outputs is

$$\mathcal{L}_{\text{recon}} = \mathbb{E}_{(\tau,s)\sim\mathcal{B}} \left[ \frac{1}{ND} \sum_{i=1}^{N} \sum_{d=1}^{D} \left( \|\hat{s}_{d,t}^i - s_{d,t}\|_2^2 + \lambda_{\text{meta}} \, \text{CE}(\hat{\xi}_{d,t}^i, \xi_{d,t}^i) \right) \right], \tag{B.2}$$

where CE denotes the cross-entropy loss and $\lambda_{\text{meta}}$ balances state reconstruction with meta-awareness in the latent representation.

To prevent unnecessary information from being encoded in the latent space, we introduce a cycle-consistency constraint. Specifically, the latent representation $z_t^i$ is reconstructed through the decoder and re-encoded using an auxiliary encoder $Enc_{c,\psi}$, ensuring that only essential state-related features remain in the latent representation. The corresponding loss is

$$\mathcal{L}_{\text{cons}} = \mathbb{E}_{(\tau,s)\sim\mathcal{B}} \left[ \frac{1}{N} \sum_{i=1}^{N} \left( \|\hat{z}_t^i - z_t^i\|_2^2 \right) \right]. \tag{B.3}$$

In parallel, policy learning is guided by the temporal-difference (TD) error, computed using the current network parameters $\theta$ and the target network parameters $\theta^-$.

$$\mathcal{L}_{\text{TD}} = \mathbb{E}_{s,a,r,s'} \left[ \left( r_t + \gamma \max_{a'} Q_{\theta^-}^{\text{tot}}(s_{t+1}, a') - Q_\theta^{\text{tot}}(s_t, a_t) \right)^2 \right]. \tag{B.4}$$

---

**Algorithm 1** LLM-driven Multi Agent Communication (LMAC)

---

1: **Initialize:** task prompt $x = (\mathcal{I}_T, \mathcal{I}_P)$, reasoning tokens $z^{(0)}$ (via CoT essential state selection), $\phi$, $\psi$, Q network
2: **for** $k = 0, 1, 2$ **do**
3:     Generate communication protocol $f_C^{(k)} \sim f_\theta^{\text{LLM}}(x, z^{(k)})$
4:     **if** $k = 0$ **then**
5:         Collect trajectory dataset $\mathcal{B}$
6:     **end if**
7:     Train discriminator $D_\phi^{(k)}$ on $\mathcal{B}$ by minimizing equation B.1
8:     Compute meta-information $\chi_{l,d,t}^{i,(k)}$
9:     Derive feedback instruction $\tilde{x}^{(k+1)}$ and generate feedback $c^{(k+1)}$
10:     Update tokens $z^{(k+1)} \sim f_\theta^{\text{LLM}}(x, c^{(k+1)})$
11: **end for**
12: Final protocol $f_C = (f_C^{(0)}, f_C^{(1)}, f_C^{(2)})$
13: **for** each training episode **do**
14:     Obtain messages $m_t^i = f_C(\tau_t)$
15:     Encode latent $z_t^i = \text{Enc}_\psi(\tau_t^i, m_t^i)$
16:     Decode to predict state $\hat{s}_t^i$ and meta-info $\hat{\xi}_{d,t}^i$
17:     Compute $\mathcal{L}_{\text{recon}}$ using Equation B.2
18:     Compute $\mathcal{L}_{\text{cons}}$ by Equation B.3
19:     Update parameters $\psi$ and by minimizing the overall objective with TD-loss for QMIX
20: **end for**

---

Finally, representation learning is jointly optimized with the TD error, and the complete objective is defined as

$$\mathcal{L} = \mathcal{L}_{\text{TD}} + \mathcal{L}_{\text{recon}} + \lambda_{\text{cons}} \mathcal{L}_{\text{cons}}. \tag{B.5}$$

The overall training procedure of LMAC is summarized in Algorithm 1.

## C  EXPERIMENTAL DETAILS

All baseline algorithms are evaluated using the official implementations and default settings released by their respective authors. The implementation of LMAC is based on EPyMARL[1] (Papoudakis et al., 2021b) , and all experiments in SMAC were conducted using StarCraft II version 2.4.10. Our method and comparisons are trained on an NVIDIA RTX 4090 GPU with an AMD EPYC 9334 CPU (Ubuntu 20.04). In the following sections, we provide details of the environments, baseline algorithms, reconstruction threshold settings, and hyper-parameter configurations used in our experiments. In particular,  C.1 outlines the environment settings,  C.2 describes the baseline algorithms, and  C.3 summarizes the hyper-parameter configurations of our implementation.

### C.1  ENVIRONMENT DETAILS

#### C.1.1  STARCRAFT MULTI-AGENT CHALLENGE WITH COMMUNICATION

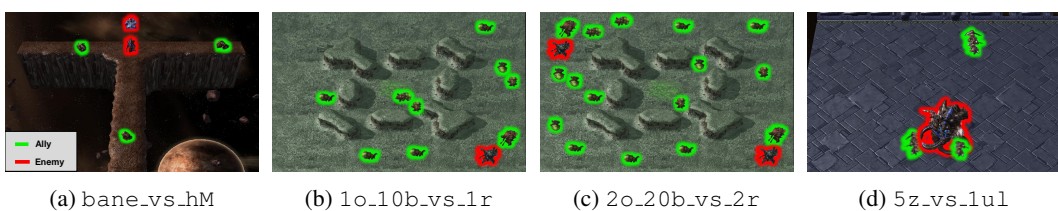

(a) bane_vs_hM        (b) 1o_10b_vs_1r        (c) 2o_20b_vs_2r        (d) 5z_vs_1ul

Figure C.1: SMAC-comm scenarios: (a) bane_vs_hM, (b) 1o_10b_vs_1r, (c) 2o_20b_vs_2r, (d) 5z_vs_1ul.

We evaluate our method on four scenarios from the StarCraft Multi-Agent Challenge with communication (SMAC-Comm). Among them, bane_vs_hM, 1o_10b_vs_1r, and 5z_vs_1ul are introduced by  Wang et al. (2020), while 2o_20b_vs_2r is a new map we propose based on 1o_10b_vs_1r. The illustrations of these scenarios are shown in Figure C.1, and their detailed configurations are summarized in Table C.1.

In SMAC-Comm, the **state space** contains absolute information of all units, including their positions, health, shields, energies, cooldowns, unit types, and most recent actions, while each agent's **observation space** is restricted to local information within its sight range, capturing relative positions, health, shield status, and unit types of nearby allies and enemies. The **action space** is defined as a set of discrete actions, including movement in four directions, attacks on visible enemies, special unit abilities, as well as stop and no-op commands, where no-op is used exclusively by eliminated units. The reward function is shaped by damage dealt to enemies, elimination of enemy units, and winning the scenario, and is formally defined as

$$R = \sum_{e \in \text{enemies}} \Delta\text{Health}(e) + \sum_{e \in \text{enemies}} \mathbb{I}(\text{Health}(e) = 0) \cdot \text{Reward}_{\text{death}} + \mathbb{I}(\text{win}) \cdot \text{Reward}_{\text{win}} \quad \text{(C.1)}$$

where $\Delta\text{Health}(e)$ denotes the health reduction of enemy unit $e$ during a timestep, $\mathbb{I}(\cdot)$ is an indicator function, and $\text{Reward}_{\text{death}}$ and $\text{Reward}_{\text{win}}$ are set to 10 and 200, respectively. A more detailed description of each scenario is provided below.

**bane_vs_hM:** Three Banelings attempt to take down a Hydralisk supported by a Medivac. Only when all three explode together can the Hydralisk be defeated, as any delay allows the Medivac to restore its health. To succeed, the Banelings must strike in perfect unison at the central junction of the T-shaped map, where the Hydralisk is positioned. This scenario requires agents to accurately perceive their positions and execute attacks simultaneously in order to succeed.

**1o_10b_vs_1r:** On a cliff-dense map, an Overseer locates a Roach that must be eliminated by its 10 Baneling allies to secure victory. While the Overseer and Roach appear together at a random spot, the Banelings spawn separately across the map. Under a minimal communication scheme, the Banelings remain silent, leaving the Overseer responsible for encoding its own position and transmitting it to guide the team.

---

[1]https://github.com/uoe-agents/epymarl

**2o_20b_vs_2r:**. This map is an extension of `1o_10b_vs_1r` that we propose. Similar to the original setting, the scenario is played on a cliff-dense map where 20 Banelings must eliminate 2 Roaches. Both Roaches and Banelings spawn at random locations across the map. This environment is designed to evaluate whether our proposed communication method remains effective in more complex scenarios with a larger number of agents.

**5z_vs_1ul:** This map features five Zealots controlled by the agents against one Ultralisk as the enemy. The Ultralisk has high health and strong melee attacks, requiring coordinated micro-management from the Zealots to win. The challenge emphasizes some tactics such as kiting strategies, positioning, and focus fire to maximize damage while minimizing losses.

Table C.1: Detailed description of SMAC-Comm scenarios

| Map | Ally Units | Enemy Units | State Dimension | Obs Dimension | Num. of Actions |
|---|---|---|---|---|---|
| bane_vs_hM | 3 Banelings | 1 Hydralisk, 1 Medivac | 52 | 31 | 8 |
| 1o_10b_vs_1r | 1 Overseer, 10 Banelings | 1 Roach | 148 | 85 | 7 |
| 2o_20r_vs_2r | 2 Overseers, 20 Banelings | 2 Roaches | 296 | 171 | 7 |
| 5z_vs_1ul | 5 Zealots | 1 Ultralisk | 63 | 36 | 7 |

### C.1.2 LEVEL-BASED FORAGING

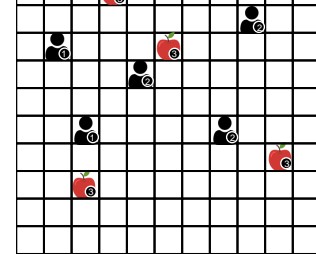

(a) `8x8_2p_2f_s1_coop`  (b) `11x11_6p_4f_s1_coop`

Figure C.2: LBF scenarios: (a) `8x8_2p_2f_s1_coop`, (b) `11x11_6p_4f_s1_coop`.

We adopt the Level-Based Foraging (LBF) variant introduced by Li et al. (2022b). The **state space** is represented as a structured grid encoding the positions and levels of all agents along with the locations and required levels of food items, rather than by concatenating individual observations. With the cooperation option enabled, each food item requires the joint effort of multiple agents, with its level set equal to the sum of the three lowest agent levels, ensuring that no agent can collect food alone and that every successful loading demands coordination. The **observation space** for each agent is limited to a $3 \times 3$ local field centered on itself, capturing relative information about nearby agents and food. The **action space** consists of six discrete actions: moving north, south, east, or west, attempting to load adjacent food, and the idle action (none). The **reward function** is cooperative and normalized by the total potential food value, granting positive returns only when the combined levels of participating agents meet or exceed the requirement of the targeted food. We evaluate two cooperative configurations as illustrated in Figure C.2.

**8x8_2p_2f_s1_coop:** A compact $8 \times 8$ grid with 2 agents and 2 food items, where cooperation is strictly enforced for every collection attempt.

**11x11_6p_4f_s1_coop:** A larger $11 \times 11$ grid with 6 agents and 4 food items under the same cooperative setting, introducing greater complexity through increased agent interactions and map size.

### C.1.3 GOOGLE FOOTBALL RESEARCH

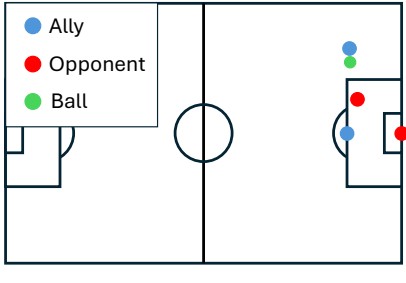
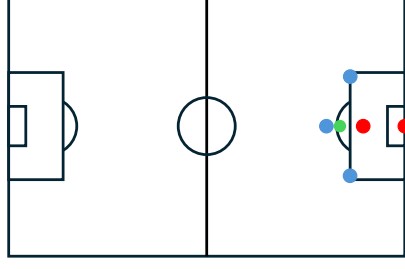

(a) `run_pass_and_shoot`          (b) `3_vs_1_with_keeper`

Figure C.3: GRF scenarios: (a) `Run_pass_and_shoot`, (b) `3_vs_1_with_keeper`.

We use the Google Research Football (GRF) environment (Kurach et al., 2020), a physics-based soccer simulator that incorporates core mechanics such as ball control, passing, shooting, tackling, and player movement. In this environment, each agent controls an individual player and must cooperate with teammates to score goals against scripted opponents. From the GRF scenarios, we consider `academy_3_vs_1_with_keeper` and `academy_run_pass_and_shoot_with_keeper`, which we refer to as `3_vs_1_with_keeper` and `run_pass_and_shoot` for brevity. The illustrations of the GRF scenarios are shown in Figure C.3, and their detailed configurations are summarized in Table C.2.

In GRF, the **state space** contains the positions and velocities of all players as well as the ball, with ally and opponent features represented in the same format. Each agent's **observation space** consists of local information about itself, nearby teammates, opponents, and ball-related features, all expressed relative to the agent's frame. The **action space** is discrete and includes movement in eight directions, sliding, passing, shooting, sprinting, and standing still, which together enable the agents to create scoring opportunities. The **reward function** is provided under two schemes: Scoring and Checkpoint. The Scoring function gives +1 for scoring a goal and -1 for conceding, while the Checkpoint function provides additional intermediate rewards such as for successful passes or defensive actions. In our experiments, we adopt the sparse Scoring function to increase the difficulty of the scenarios. A more detailed description of each scenario is provided below.

**3_vs_1_with_keeper:** Three attackers operate from the edge of the box: one on each wing and one in the center. The central player begins with the ball while directly confronted by a defender, and an opposing goalkeeper guards the net. The scenario emphasizes teamwork through passing and positioning to create scoring opportunities.

**run_pass_and_shoot:** Two attackers are positioned near the edge of the penalty area. One player starts wide with possession and is unmarked, while the other is placed centrally, marked by a defender, and facing the goalkeeper. The setup encourages passing and coordinated shooting to overcome the defense.

Table C.2: Detailed description of GRF scenarios

| Scenario | Ally | Opponent | State Dim | Obs Dim | Action Dim |
|---|---|---|---|---|---|
| `3_vs_1_with_keeper` | 3 central midfield | 1 goalkeeper, 1 center back | 26 | 26 | 19 |
| `Run_pass_and_shoot` | 2 central back | 1 goalkeeper, 1 center back | 22 | 22 | 19 |

## C.2 Detailed Description of Baseline Algorithms

**QMIX (Rashid et al., 2018)** QMIX factorizes the joint action-value function into individual utilities using a monotonic mixing network. It provides a strong baseline for cooperative MARL under centralized training with decentralized execution, but does not involve explicit communication between agents. We base our implementation of QMIX on the following repository: `https://github.com/hijkzzz/pymarl2`

**FullComm** A variant of QMIX where each agent broadcasts its full local observation to all others at every timestep. This represents an upper-bound setting with maximal communication capacity but incurs heavy redundancy and communication cost.

**QMIX+State** An oracle-like upper bound where each agent is directly given the global state in addition to its local observation. This allows agents to make fully informed decisions and serves as a reference for the maximum achievable performance.

**NDQ (Wang et al., 2020)** Neural Decomposable Q-learning introduces nearly decomposable Q-functions that minimize communication overhead. Agents act independently most of the time, but exchange messages guided by information-theoretic regularizers that maximize mutual information while minimizing entropy. This approach achieves strong coordination while reducing communication by over 80% compared to full exchange. The official code can be found at: `https://github.com/TonghanWang/NDQ`

**MASIA (Li et al., 2022b)** Multi-Agent Self-supervised Information Aggregation enables agents to aggregate received raw messages into compact, permutation-invariant representations. These embeddings are optimized through self-supervised objectives such as reconstruction and prediction, allowing agents to extract the most relevant information for decision-making and significantly improve coordination. The official code can be found at: `https://github.com/chenf-ai/Multi-Agent-Communication-Considering-Representation-Learning`

**MAIC (Du et al., 2022)** Multi-Agent Incentive Communication allows each agent to generate incentive messages that directly bias teammates' value functions, promoting explicit coordination. By learning targeted teammate models and applying sparsity regularization, MAIC improves efficiency and achieves strong performance across diverse cooperative MARL benchmarks. The official code can be found at: `https://github.com/mansicer/MAIC`

**COLA (Monda et al., 2023)** Consensus Learning for Agents enables cooperative behavior by allowing agents to infer a shared consensus representation from their local observations. Even without direct access to the global state, agents learn viewpoint-invariant representations that converge to the same discrete consensus, which is then used as an additional input for decentralized decision-making. The official code can be found at: `https://github.com/deligentfool/COLA`

**T2MAC (Liu et al., 2024)** Targeted and Trusted Multi-Agent Communication equips agents with mechanisms for selective engagement and evidence-driven message integration. Agents decide when and with whom to communicate, exchange individualized messages, and integrate received information at the evidence level, leading to more efficient and reliable cooperation. The official code can be found at: `https://github.com/ZangZehua/T2MAC`

## C.3 Hyper-parameter Setup

We calibrated the reconstruction threshold $\alpha$ according to the spatial structure and visibility conditions of each environment: 0.05 for SMAC-Comm where normalized coordinates directly reflect spatial error, a stricter 0.005 for the more challenging `bane_vs_hM` scenario, 0.002 for GRF with absolute field coordinates and weaker observability limits, and 0.1 for the grid-based LBF where prediction depends on cell occupancy. Beyond these thresholds, default hyper-parameters were used as the baseline configuration. For each scenario, we primarily followed the settings provided by the original authors; when such specifications were unavailable, the default parameters were applied. The full set of hyper-parameters used in our experiments is summarized in Table C.3.

Table C.3: Common hyper-parameter setting of LMAC

| Parameter | Value |
| --- | --- |
| Hidden dimension for self-attention module | 64 |
| Latent dimension | 20 |
| Dropout rate | 0.1 |
| Optimizer | Adam |
| $\epsilon$ anneal step | 50000 |
| $\epsilon$ Decay Value | $1.0 \rightarrow 0.05$ |
| Replay buffer size | 5000 |
| Target update interval | 200 |
| Mini-batch size | 32 |
| Mixing network dimension | 32 |
| Discount factor $\gamma$ | 0.99 |
| Learning rate | 0.0005 |
| Coefficient of meta-information loss $\lambda_{\text{meta}}$ | 0.1 |
| Coefficient of consistency loss $\lambda_{\text{cons}}$ | 1 |
| temperature(LLM generation) | 0.6 |

# D   ADDITIONAL TRAJECTORY ANALYSIS

In addition to the main trajectory analysis presented in the paper, we further examine protocol refinement in other SMAC-Comm scenarios and GRF tasks. These supplementary cases demonstrate that the framework yields interpretable communication protocols whose iterative refinements adapt to scenario-specific challenges. The following analyses provide examples from different environments, illustrating how the protocol evolves beyond the scenarios presented in Section 5.2.

**SMAC-comm: bane_vs_hM**

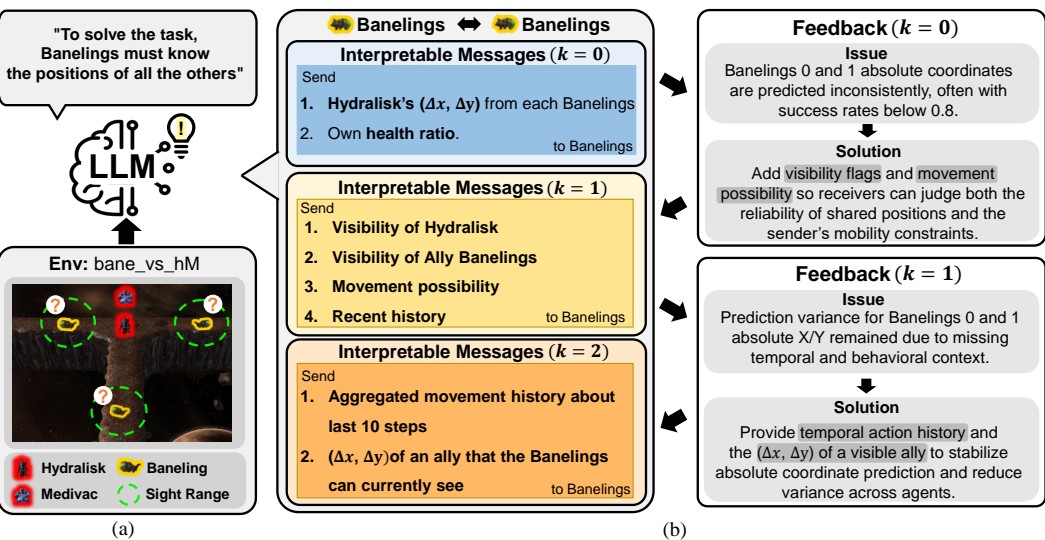

(a)                                                                          (b)

Figure D.1: Protocol refinement analysis on SMAC bane_vs_hM: (a) Task scenario with Banelings, Hydralisk and Medivac under partial observability, (b) protocol messages and corresponding feedback at each phase $k$

As a complementary case, we analyze protocol refinement in the SMAC bane_vs_hM map, summarized in Fig. D.1. In (a), three Banelings must coordinate a simultaneous detonation against a Hydralisk supported by a Medivac, where precise synchronization is critical. Because absolute coordinates are absent from local observations and the long vertical corridor makes $y$-position inference particularly difficult, agents struggle to align their attacks without additional cues. In (b), protocol evolution is shown: at $k = 0$, Banelings broadcast the Hydralisk's relative position $(\Delta x, \Delta y)$ and their own health ratio, which provides partial but unreliable signals, resulting in inconsistent absolute localization. Feedback highlights this instability and suggests including visibility indicators, movement possibilities, and recent history. At $k = 1$, these additions improve the interpretability of shared information, but prediction variance remains high for certain coordinates due to missing temporal and behavioral context. At $k = 2$, variance-based feedback leads to incorporating aggregated movement history over the last 10 steps together with the relative position of currently visible allies, allowing agents to stabilize absolute predictions and achieve consistent coordination. These results show that refining protocols to generate and share structured temporal-behavioral features, rather than only raw observations, is key to enabling consistent absolute localization under partial observability.

**GRF: Run_pass_and_shoot**

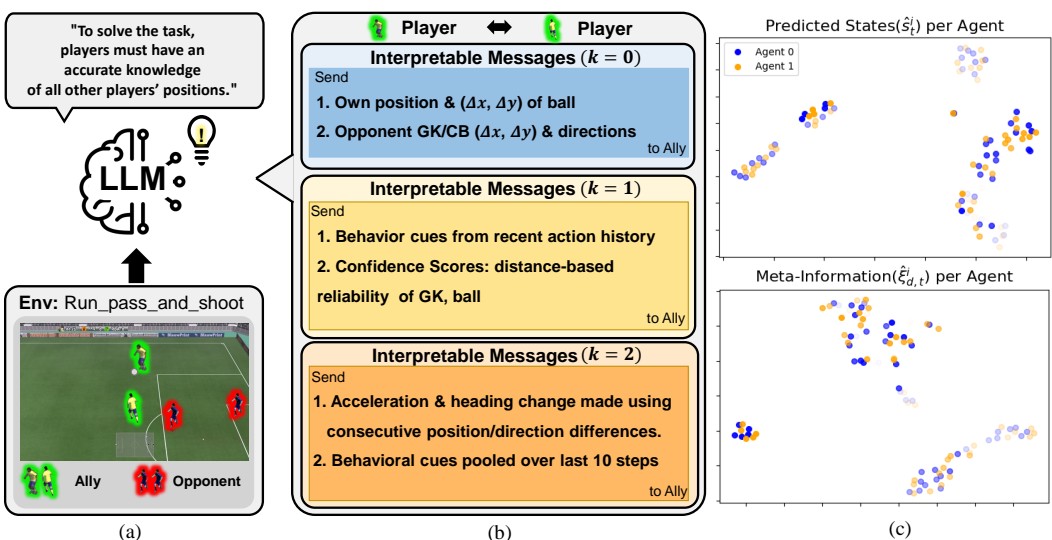

(a)          (b)          (c)

Figure D.2: Protocol refinement analysis on GRF `Run_pass_and_shoot`: (a) Task scenario with two attackers, a defender, and goalkeeper near the penalty area, (b) protocol messages at each phase $k$, (c) t-SNE of predicted states and meta-information showing convergence across agents after communication.

As a complementary case, we analyze protocol refinement in the GRF `Run_pass_and_shoot` scenario, summarized in Fig. D.2. In (a), two attackers must cooperate near the penalty area against a central defender and a goalkeeper. Although the state space in GRF is structurally simpler than in SMAC, it remains important that agents infer states from shared messages and incorporate them into policy decisions, a pattern that is also observed in LBF. (b) shows the protocol evolution. At $k = 0$, each agent shares its own position, the relative displacement of the ball, and the positions of the central defender and goalkeeper, but such information alone is limited for predicting other aspects of the state. Accordingly, at $k = 1$, behavioral cues such as pass/shoot readiness and sprinting, together with confidence scores regarding the goalkeeper and ball, are added. At $k = 2$, dynamic features such as acceleration and heading changes, along with aggregated behavioral histories over the last 10 steps, are incorporated, stabilizing predictions and enabling cooperative play in which the wide attacker penetrates open space while the central striker draws defensive pressure. In (c), the t-SNE visualization shows that the predicted states $\hat{s}t^i$ and the meta-information $\hat{\xi}d, t^i$ converge across agents after communication. This indicates that through message exchange, all agents come to predict state dimensions at a similar level and, moreover, share a common recognizability of which dimensions are reliably captured.

# E ADDITIONAL EXPERIMENTAL ANALYSES

In this section, we present additional experimental analyses to further examine LMAC. We first assess the generality of LMAC by combining it with different value decomposition methods and scaling to larger environments with more agents, as shown in E.1. We then explore how performance changes when the feedback-based refinement is applied multiple times, as detailed in E.2. Lastly, we study the influence of restricting the message dimension on protocol design and communication efficiency, which is discussed in E.3.

## E.1 GENERALITY OF LMAC

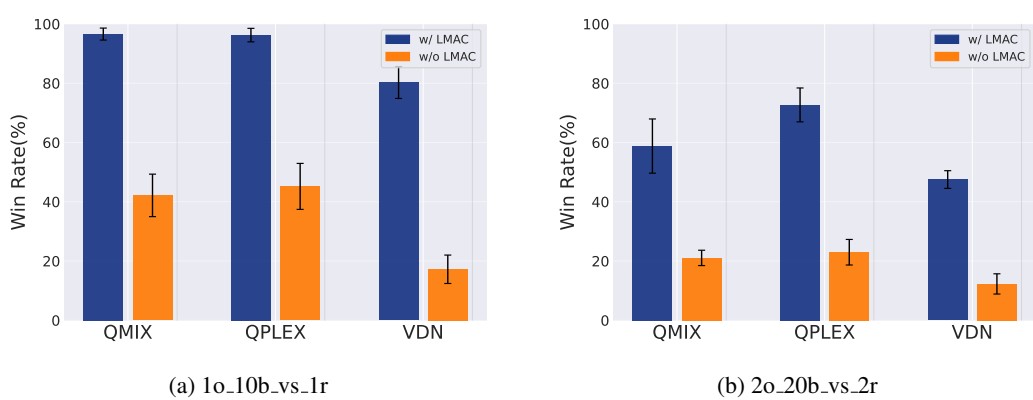

(a) 1o_10b_vs_1r
(b) 2o_20b_vs_2r

Figure E.1: Generality of LMAC across different value decomposition methods (VDN, QMIX, QPLEX) and larger environments with more agents

We further evaluate the generality of LMAC by combining it with different value decomposition algorithms (VDN, QMIX, QPLEX). As shown in Figure E.1, LMAC provides a consistent performance boost across all methods, not only when paired with QMIX. In addition, this benefit is preserved in more complex environments with larger numbers of agents, such as 2o_20b_vs_2r, demonstrating that the effectiveness of LMAC scales beyond simple scenarios. These results confirm that the proposed communication framework generalizes well across both algorithmic backbones and environmental complexities.

## E.2 EFFECT OF THE NUMBER OF UPDATE PHASES

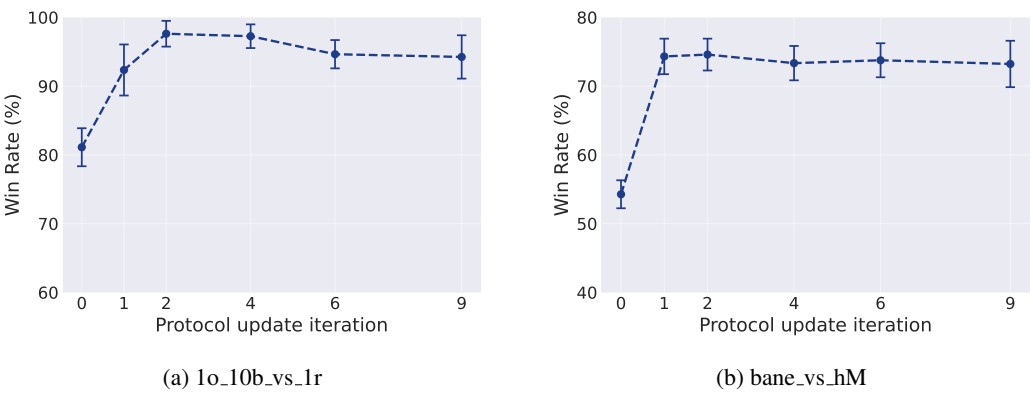

(a) 1o_10b_vs_1r
(b) bane_vs_hM

Figure E.2: Performance comparison when the communication protocol is iteratively updated across different numbers of update phases in (a) 1o_10b_vs_1r and (b) bane_vs_hM.

We analyze the effect of repeatedly applying feedback-based protocol refinement using the Sharing Enhancement update scheme. As shown in Figure E.2, performance improves as the number of update phases $k$ increases, but the marginal gain quickly saturates around $k = 3$. In fact, even $k = 2$ is sufficient to capture most important state dimensions that can be inferred from observations, while larger $k$ mainly increases message size and introduces redundant information, reducing efficiency. Nevertheless, in environments that demand more sophisticated reasoning, employing more refinement phases may still offer benefits.

### E.3 EFFECT OF REDUCED COMMUNICATION CAPACITY UNDER CONSTRAINTS

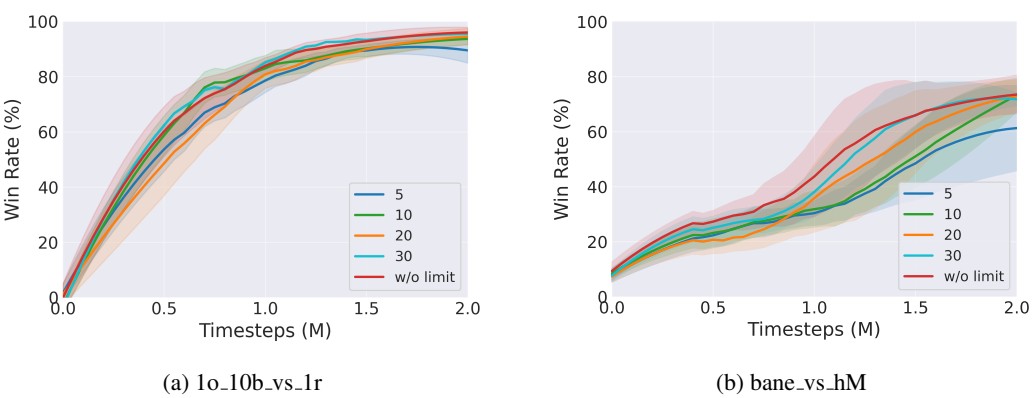

(a) 1o_10b_vs_1r

(b) bane_vs_hM

Figure E.3: Performance comparison under message dimension constraints in (a) `1o_10b_vs_1r` and (b) `bane_vs_hM`.

We further investigated how performance changes when message dimensionality is constrained, since larger message sizes naturally lead to higher communication overhead. Here, we use communication capacity to denote the effective amount of information that agents can transmit through messages, which is directly determined by message dimensionality. Thus, restricting the number of message dimensions can be regarded as limiting the communication capacity of agents. As shown in Fig. E.3, our method remains robust under such conditions: even with reduced message sizes, performance is largely preserved. In particular, the LLM reduced overhead by designing protocols that avoid unnecessary all-to-all communication through one-way broadcast structures, or by compactly encoding key features such as movement possibility, last action, and sender ID into only a few bits. However, in more challenging scenarios such as `bane_vs_hM`, where state inference is inherently more difficult, excessive compression slowed convergence, indicating that a moderate level of communication capacity is still necessary for effective learning.

# F COMPARISON OF COMPUTATIONAL COMPLEXITY

In the SMAC communication experiments, we measured the total training time for 2M steps on the `bane_vs_hM` and `1o_10b_vs_1r` maps, as reported in Table F.1. On average, LMAC requires about 15% more training time than strong baselines such as MASIA and MAIC. This overhead mainly comes from training the discriminator and collecting additional trajectory data for it, but it is a necessary cost that allows the model to diagnose weaknesses in the communication protocol and iteratively refine it. As a result, LMAC consistently achieves higher performance than all baselines, demonstrating that the improvement in coordination quality outweighs the extra computation.

Table F.1: Total training time (hours) for 2M steps in SMAC communication settings.

| Algorithm | bane_vs_hM | 1o_10b_vs_1r |
|-----------|------------|--------------|
| QMIX | 4h 12m | 5h 42m |
| NDQ | 5h 17m | 6h 33m |
| T2MAC | 5h 34m | 7h 28m |
| MASIA | 7h 13m | 8h 13m |
| MAIC | 7h 45m | 8h 32m |
| COLA | 7h 54m | 8h 24m |
| LMAC(Ours) | 8h 26m | 9h 47m |

