# OpenReview forum: "LLM-Guided Communication for Cooperative Multi-Agent Reinforcement Learning"
_ICLR.cc/2026/Conference — ICLR 2026 Conference Withdrawn Submission_

### Official Review · Reviewer_JdpQ · 2025-10-26

**Soundness:** 2
**Presentation:** 3
**Contribution:** 2
**Rating:** 2
**Confidence:** 4

**Summary:**

This paper proposes LMAC (LLM-driven Multi-Agent Communication), a novel framework that leverages Large Language Models (LLMs) to design interpretable, task-specific communication protocols for cooperative Multi-Agent Reinforcement Learning (MARL). Also, the paper adopts a latent module to process the messages for the final decision-making. LMAC is evaluated on classic MARL benchmarks - SMAC-Comm, Level-Based Foraging, and Google Research Football.

**Strengths:**

1. Comprehensive empirical evaluation across three families of MARL benchmarks with multiple scenarios and seeds.
2. Clear presentation with detailed methodological descriptions, prompts, and appendices. The trajectory-level analyses and phase-wise meta-information provide useful qualitative insight into how messages evolve.
3. Broad baseline coverage within the communication literature (QMIX, FullComm, QMIX+State, NDQ, MASIA, MAIC, COLA, T2MAC), and ablations on phases, consistency loss, meta-information, LLM variants, and reconstruction thresholds.

**Weaknesses:**

1. Limited novelty. The core idea—asking an LLM to draft and refine communication rules and then encoding those messages with a standard autoencoder plus cycle consistency—is incremental. Much of the gain appears to come from adding a stronger representation module with the help from a giant LLM rather than a fundamentally new learning principle for communication.
2. Missing analysis of token and computation costs. The paper does not report LLM token usage, number of queries per phase, wall-clock overhead, and API cost for protocol refinement. This is important for assessing real-world feasibility.
3. Credibility of the LLM-driven piece is not thoroughly established. The work assumes that LLM-derived protocols are needed and better than learned protocols, but it does not include a controlled baseline where the same latent module is paired without LLM prompts.
4. Underutilization of LLM Capabilities: The paper's framing raises a fundamental question about the appropriate level of abstraction for leveraging an LLM. If an LLM has the advanced reasoning capacity to understand complex environment dynamics, state-action spaces, and multi-agent objectives to design a communication protocol, it is arguably capable of contributing to more critical, higher-level tasks. For example, it could be used for strategic planning, sub-goal decomposition, or even direct policy guidance. Limiting the LLM's role to generating a static communication function seems like an underutilization of its potential and suggests that the problem could be framed in a more impactful way.

**Questions:**

1. Do the protocols generalize to unseen scenarios (new maps or different observation encodings) without re-prompting the LLM?
2. Tough it may beyond the scope of this paper, can LLMs improve decision making? For example, can the LLM generate macro-action policies, role assignments, or planning templates, and how do these compare with communication-only LMAC?
3. How does the latent module specifically designed for LLM-guided messages?

---

### Official Review · Reviewer_XxUX · 2025-10-27

**Soundness:** 3
**Presentation:** 2
**Contribution:** 2
**Rating:** 2
**Confidence:** 5

**Summary:**

LMAC proposes to (i) use an LLM to synthesize an interpretable, task‑specific communication protocol via three refinement phases—Protocol Initialization → Recognition Enhancement → Sharing Enhancement—guided by a learned discriminator that turns trajectory data into dimension‑wise “can this be recovered?” meta‑signals; and (ii) plug these messages into CTDE training through a meta‑cognitive latent module that performs state reconstruction, attaches reliability signals, and imposes cycle consistency to strip redundancy (see Fig. 1, p. 2; Fig. 2, pp. 4–5; Fig. 4, p. 6). Empirically, LMAC improves over communication baselines on SMAC‑Comm, LBF, GRF, converging faster and even surpassing QMIX+State on GRF (see Fig. 6, p. 7). Ablations show each phase and each loss term matters; results are fairly robust to the reconstruction threshold and LLM choice (see Table 1, p. 9).

**Strengths:**

The paper reframes protocol learning as interpretable program synthesis with phase‑specific, Reflexion‑style feedback that is grounded in empirical reconstruction statistics; this is a crisp way to force messages to carry task‑critical, consistently understood content. The meta‑cognitive representation—predicting both state and “which dimensions I can recover reliably”—plus cycle consistency yields compact, stable features that integrate cleanly with QMIX/VDN/QPLEX under CTDE (see Fig. 4, p. 6; Eq. (B.5), p. 22). Experiments span three families of benchmarks with careful phase‑wise analyses (e.g., Fig. 7, p. 8) and solid ablations (Table 1, p. 9), and the method scales to a larger SMAC‑style scenario (2o_20b_vs_2r, Appx. C/E, pp. 24–25, 30). Overall presentation is clear, with prompts, pseudo‑code, and hyperparameters provided (Appx. B/C), aiding reproducibility.

**Weaknesses:**

1.System complexity and compute overhead.
LMAC couples an LLM‑generated protocol, a discriminator, and a latent encoder–decoder. The paper acknowledges extra cost (≈15% wall‑time vs. strong comm baselines on SMAC‑Comm; see Table F.1, p. 32) but this trade‑off deserves a more dialectical discussion in the main text: where do the wins come from vs. which parts add cost; how does cost scale with more agents or more phases; and when is the overhead unjustified (e.g., low‑dimensional states). Moving a compute‑efficiency analysis from Appendix F to the main body would strengthen the narrative.

2.Reference accuracy and bibliography hygiene.
Some citations look inconsistent. The Related‑Work section lists both “TMC: Targeted multi-agent communication” and “TarMAC: Targeted multi‑agent communication” as separate entries by the same authors/year (Das et al., 2019a/2019b), which suggests a possible duplicate or naming confusion that should be reconciled. Likewise, please double‑check the formatting and venues for MAIC and MASIA and ensure the in‑text mentions line up with the bibliography entries (Sec. 3, pp. 3–4; Refs., pp. 10–14).

3.Comparative fairness with stronger recent baselines.
Most baselines are reputable (NDQ’20, MASIA’22, MAIC’22, COLA’23, T2MAC’24), but if the paper’s main thrust is efficiency and interpretability, it would be fairer to add modern, efficiency‑oriented or interpretable comm methods beyond those already included (e.g., stronger gating/scheduling, robustness‑to‑noise, or semantics‑focused communication), and to standardize training budgets and model sizes across methods. At minimum, add a brief “why these baselines” rationale in Sec. 5.1 (p. 7).

4.Terminology and attributions (SMAC‑Comm).
The main text introduces SMAC‑Comm as “SMAC with communication” and attributes specific maps to Samvelyan’19 and Wang’20, while 2o_20b_vs_2r is explicitly introduced by the authors (Appx. C.1, pp. 23–24). That is acceptable, but the paper should be explicit up front that “SMAC‑Comm” is a composite setup (existing SMAC variants plus one new scenario), to avoid the impression that a third‑party benchmark by that exact name exists. Clarify this in Sec. 5 (p. 6) and Appx. C.1.

5.Editing/consistency issues you should fix.
– Fig. 5 caption says “MALR Benchmarks” (typo for MARL, p. 6).
– Table C.1 row uses “2o 20r vs 2r” (likely a typo for 2o 20b vs 2r, p. 24).
– Minor notation drift exists around α settings between main text and Appx. C.3 (pp. 26–27).
Cleaning these will improve polish.

**Questions:**

Q1. Many‑agent, heterogeneous settings; how about SMAC‑V2?
The paper does test a larger agent count with 2o_20b_vs_2r and reports that LMAC’s advantage grows when agents scale (see Fig. 6, p. 7, and the generality test in Fig. E.1, p. 30). There are also heterogeneous‑unit scenarios (e.g., bane_vs_hM, 5z_vs_1ul), but SMAC‑V2 is not evaluated, and the heterogeneity is moderate. Thus, strong results for SMAC‑V2 cannot be claimed. A reasonable expectation is that LMAC’s phase‑wise refinement and meta‑signals will help when state recovery is the bottleneck, but long‑horizon credit assignment and severe non‑stationarity in SMAC‑V2 could stress the discriminator and inflate α‑sensitivity. The authors should add SMAC‑V2 or at least a synthetic many‑agent heterogeneous stress test to validate scalability.

Q2. Should the paper include a toy environment to explain why LMAC works?
Yes. The paper’s phase‑wise plots (e.g., Fig. 3, p. 6; Fig. 7, p. 8; Appx. D/E) are helpful, but a minimal toy would make the mechanism undeniable. A good choice is a triangulation gridworld: N agents on a 2‑D torus must localize a hidden target with only relative bearings, where the optimal protocol needs (i) a shared anchor, (ii) per‑agent visibility flags, (iii) short‑horizon motion history. This setting would show: k = 0 finds relative deltas; k = 1 adds anchors/history to raise average recovery; k = 2 adds IDs/reliability to reduce inter‑agent variance—the exact patterns emphasized in Fig. 7(c–d), p. 8.

Q3. If an LLM is available, why not let it act as the policy?
The paper’s design deliberately keeps the policy as a learned RL function while using the LLM only once per phase to synthesize a static, verifiable protocol (see Alg. 1, p. 22). This has several advantages:

Sample‑efficiency & stability: policies still benefit from off‑policy TD learning, target networks, and value factorization under CTDE.

Determinism & latency: the protocol becomes a piece of code with no runtime LLM calls; actions are fast and hardware‑agnostic.

Safety & debuggability: messages are inspectable tensors; the discriminator provides measurable feedback on what was recovered and how consistently (Eq. (3), p. 5; Fig. 3, p. 6).
End‑to‑end LLM policies (text‑in‑the‑loop at every step) could work, but introduce inference latency, cost, and non‑determinism; the authors chose a hybrid that preserves RL stability while harvesting LLM reasoning where it helps most—protocol design.

Q4. Applicability to embodied / multi‑robot settings.
The method is agnostic to the low‑level simulator and already handles reduced comm capacity well (performance degrades gracefully when message dimensions are capped; Fig. E.3, p. 31). For physical robots you would need: (i) channel‑noise‑aware meta‑signals (e.g., learn α as a calibrated confidence, not a hard threshold), (ii) asynchronous timestamps in the protocol (already hinted by temporal cues in Appx. B.2, pp. 20–21), and (iii) latency‑robust anchors (global frame or UWB beacons). None of these is conceptually at odds with LMAC’s pipeline; the missing piece is an empirical validation on a real robot stack.

Q5. Limitations and future work (beyond what the paper lists).
The paper lists overhead and LLM‑dependence as limitations (Sec. 6, p. 9; Appx. F, p. 32). Additional items worth stating upfront:

Theory: no information‑theoretic optimality of the protocol or a bound connecting α, message size, and team value improvement.

Generalization: no evaluation on SMAC‑V2 or ad‑hoc teamwork; protocol portability across tasks is untested.

Robustness: resilience to noisy/lossey channels and adversarial messaging is not studied; a reliability‑aware loss that penalizes over‑confident meta‑signals would help.

Open‑world adaptation: an online refresh schedule (e.g., small buffer + lightweight D update + 1‑shot Sharing‑Enhancement) is a practical extension.

Human‑facing interpretability metrics: beyond visualizations (Fig. 7; Appx. D), add third‑party ratings or task‑grounded comprehension tests.

Typos：
Note on issues you asked me to re‑check:

“MALR”→MARL typo (Fig. 5 caption, p. 6).

“2o 20r vs 2r”→2o 20b vs 2r typo (Table C.1, p. 24).

Potential TMC/TarMAC duplication in references (Sec. 3 and Refs., pp. 3–4, 10–11).

Baselines and hyperparameters are indeed in Appx. C, with hardware and LLM variants specified (pp. 23, 26–27; Table 1(b), p. 9).

Ref:

[1]A survey of progress on cooperative multi-agent reinforcement learning in open environment

[2]Multi-agent embodied ai: Advances and future directions

[3]Language Grounded Multi-agent Reinforcement Learning with Human-interpretable Communication

---

### Official Review · Reviewer_vvtw · 2025-11-01

**Soundness:** 3
**Presentation:** 3
**Contribution:** 3
**Rating:** 4
**Confidence:** 4

**Summary:**

This paper presents LMAC, a multi-agent reinforcement learning framework that leverages large language models (LLMs) to generate and refine communication protocols through three phases: Protocol Initialization, Recognition Enhancement, and Sharing Enhancement.

Each phase employs a discriminator to provide feedback on communication effectiveness, enabling iterative protocol refinement. A meta-cognitive latent module is introduced to reconstruct global states with dimension-wise recovery signals and enforce cycle-consistency, ensuring compact and interpretable message design.

Experiments on SMAC-Comm, LBF, and GRF benchmarks demonstrate that LMAC consistently outperforms prior methods in terms of information recovery, inter-agent consistency, and cooperative performance

**Strengths:**

1.The approach is technically sound, combining reconstruction and consistency losses to ensure stable and interpretable communication. Experiments on SMAC-Comm, LBF, and GRF are comprehensive and clearly presented. Figures and appendices with protocol evolution and prompt templates enhance readability and reproducibility.

2.The work extends the reasoning capability of LLMs to interpretable multi-agent communication design, offering a new paradigm for language-guided cooperative intelligence with strong potential impact on future MARL research.

**Weaknesses:**

1.While the application of LLMs to generate interpretable natural language messages for multi-agent communication is a promising direction, to the best of my knowledge, it is not the first work to do so. The authors  highlight two key aspects of their contribution（1）existing work remains confined to simple tasks and (2)largely imitates LLM agents rather than ensuring balanced situation awareness.  I find the first point well-justified and valid. However, the second point is somewhat unclear.

2.The incorporation of self-supervised auxiliary tasks like reconstruction for strategy refinement is quite conventional in multi-agent communication.

3.The Limitation section briefly states that “the discriminator cost is small and the number of LLM queries per phase is limited,” but it lacks per-benchmark and per-phase quantitative reporting. Although Appendix F provides training time comparisons with other communication baselines, it remains unclear whether these times include the LLM inference and waiting overhead. If the reported training cost only accounts for the RL component but excludes the latency of LLM calls, the comparison may not be entirely fair.

4.In Figure 6, there is an inconsistency between the legend and the plotted curves:
the color assigned to T2MAC in the legend does not match its actual curve color, and in some subplots (e.g., GRF or LBF), the win-rate curve for T2MAC is missing, yet no explanation is provided in the text.

**Questions:**

1.In Equation (3), χ determines “successful reconstruction” based on a threshold α. Could the authors explain why a binary formulation was chosen instead of a continuous reconstruction metric? Additionally, is α fixed across different tasks, or was it tuned per environment?

2.Appendix B.2 mentions using Reflexion to correct errors in the LLM-generated communication protocols. On average, how many correction rounds are required to obtain an executable protocol? Do the authors have any statistics on the failure rate or the common types of generation errors encountered?

3.The meta-information was designed to quantify how communication improves state reconstruction and inter-agent consistency, but could there be redundancy in its definition? For instance, certain dimensions may be successfully reconstructed yet irrelevant to decision-making, while still contributing to the meta-information metric.

---

### Official Review · Reviewer_kW46 · 2025-11-01

**Soundness:** 2
**Presentation:** 2
**Contribution:** 2
**Rating:** 4
**Confidence:** 5

**Summary:**

Communication is central to cooperative MARL and calls for interpretable protocols that yield consistent state understanding. The paper proposes a communication-centric framework integrating LLM-guided protocol refinement with metacognitive latent representation learning; the LLM designs protocols in three stages (initialization, recognition enhancement, sharing enhancement), and the metacognitive module injects them into CTDE via state reconstruction, dimension-wise reliability cues, and cycle consistency. Yet the interpretability claim lacks rigorous substantiation, and the experiments are inadequate.

**Strengths:**

1. The method designed by the author is logical and clear.
2. The paper provides analyses of bandwidth sensitivity and reports moderate training-time overheads.
3. Implementation details for prompts/templates improve readability and potential reproducibility.

**Weaknesses:**

1. Although it encompasses many related works, it is still insufficient.
2. The scope of interpretability extends beyond the author's research objectives.
3. More experiments are needed.
4. The newly introduced custom scenario is not yet accompanied by full release scripts, which may pose a risk to reproducibility.

**Questions:**

1. When calculating costs in the report, the analysis only considered GPU training time and did not account for the token costs associated with LLM queries.
2. The training of the discriminator relies on the global state of offline data. Without this data, how should it be handled? Is it still effective?
3. The novelty of the method has been overstated; it essentially involves using an LLM to formulate a heuristic communication strategy.

---

### Note · Authors · 2025-11-20

I have read and agree with the venue's withdrawal policy on behalf of myself and my co-authors.